# Bacterial protoplast-derived nanovesicles carrying CRISPR-Cas9 tools re-educate tumor-associated macrophages for enhanced cancer immunotherapy

Mingming Zhao[1,3], Xiaohui Cheng[1,3], Pingwen Shao[1], Yao Dong[1], Yongjie Wu[1], Lin Xiao[1], Zhiying Cui[1], Xuedi Sun[1], Chuancheng Gao[1], Jiangning Chen [1,2] ✉, Zhen Huang [1] ✉ & Junfeng Zhang [1] ✉

The CRISPR-Cas9 system offers substantial potential for cancer therapy by enabling precise manipulation of key genes involved in tumorigenesis and immune response. Despite its promise, the system faces critical challenges, including the preservation of cell viability post-editing and ensuring safe in vivo delivery. To address these issues, this study develops an in vivo CRISPR-Cas9 system targeting tumor-associated macrophages (TAMs). We employ bacterial protoplast-derived nanovesicles (NVs) modified with pH-responsive PEG-conjugated phospholipid derivatives and galactosamine-conjugated phospholipid derivatives tailored for TAM targeting. Utilizing plasmid-transformed *E. coli* protoplasts as production platforms, we successfully load NVs with two key components: a Cas9-sgRNA ribonucleoprotein targeting *Pik3cg*, a pivotal molecular switch of macrophage polarization, and bacterial CpG-rich DNA fragments, acting as potent TLR9 ligands. This NV-based, self-assembly approach shows promise for scalable clinical production. Our strategy remodels the tumor microenvironment by stabilizing an M1-like phenotype in TAMs, thus inhibiting tumor growth in female mice. This in vivo CRISPR-Cas9 technology opens avenues for cancer immunotherapy, overcoming challenges related to cell viability and safe, precise in vivo delivery.

The CRISPR-Cas9 genome editing system offers significant potential in cancer therapy, owing to its precision in targeting key oncogenes and tumor suppressor[1,2]. However, current clinical trials employing CRISPR-Cas9 for cancer treatment are primarily focused on isolating autologous T cell from patients, subjecting them to CRISPR-Cas9-mediated gene editing, and subsequently reinfusing them into the patients[3,4]. Safely and effectively manipulating specific genomic sequences within the tumor microenvironment remains a significant

challenge for the clinical application of CRISPR-Cas9 in cancer treatment[5,6]. To achieve a satisfactory therapeutic effect against cancer, CRISPR-Cas9 therapy must address the important concern about the fitness of edited cells in vivo. Oncogenes and cell death-related genes of tumor cells are often selected as the editing targets for CRISPR-Cas9 therapy[7]. The edited tumor cells typically exhibit reduced resilience, decreased proliferation and increased susceptibility to apoptosis. This may lead to the unedited tumor cells in tumor

[1]State Key Laboratory of Pharmaceutical Biotechnology, School of Life Sciences, Nanjing University, Nanjing, Jiangsu 210023, China. [2]State Key Laboratory of Analytical Chemistry for Life Sciences, Nanjing University, Nanjing, Jiangsu 210023, China. [3]These authors contributed equally: Mingming Zhao, Xiaohui Cheng. ✉e-mail: jnchen@nju.edu.cn; zhenhuang@nju.edu.cn; jfzhang@nju.edu.cn

microenvironment rapidly taking over tumor tissue and ultimately reducing the anticancer efficacy mediated by gene-editing tools[8]. To overcome this challenge, tumor-associated macrophages (TAMs) present an ideal target for gene editing. Using CRISPR-Cas9, several relevant genes can be knocked out to permanently reshape TAMs into an anti-tumor M1-like phenotype, while preserving their fitness[9,10]. These phenotype-stabilized macrophages can persistently exert anti-tumor effects without succumbing to the immunosuppressive tumor microenvironment, maximizing the efficacy of gene editing therapies. Therefore, apart from the tumor cells in the tumor microenvironment, TAMs represent a promising target for enhancing the efficacy of gene editing therapies against cancer.

Effective delivery carriers are crucial for the successful application of the CRISPR-Cas9 genome editing system in vivo, as they must accurately transfer the system into the nucleus of targeted cells[2,5]. Currently, there are three models of genome editing tools available: the plasmid-based CRISPR-Cas9 system, the sgRNA/Cas9 mRNA mixture, and the Cas9-sgRNA ribonucleoprotein (RNP)[6]. Among these models, Cas9-sgRNA RNP delivery is advantageous due to its rapid editing, reduced off-target effects, and weak immune responses. However, the high molecular weight and easy degradation of the Cas9-sgRNA RNP hinder its in vivo application through conventional drug delivery systems[11]. Recently, bacteria-derived vesicles have emerged as promising drug delivery systems. Bacteria can be genetically modified to endow vesicles (e.g. outer membrane vesicles, OMVs) with in vivo targeting properties and the ability to encapsulate endogenously expressed therapeutic agents[12,13]. However, significant hurdles regarding their endotoxicity, low yield, and uncertainty in loading components must be addressed to fully realize the clinical potential of bacteria-derived vehicles[14–16].

In this study, we develop a gene-editing delivery system using functionalized nanovesicles (NVs) derived from E. coli protoplasts to effectively encapsulate Cas9-sgRNA RNP. This approach enables the reprogramming of tumor-associated macrophages (TAMs) and demonstrates efficacy in in vivo tumor treatment. We use E. coli transformed with a plasmid encoding the Cas9-sgRNA RNP, serving as both the source of gene editing tools and the raw material for carrier preparation, simplifying the construction process of delivery carriers (Fig. 1). Furthermore, lysozyme treatment eliminates toxic outer membrane and periplasmic components, reducing protoplast virulence. Physical extrusion of the protoplasts ensures that the vesicles inherit the contents from the parent bacteria, resulting in a higher vesicle yield compared to naturally occurring bacterial vesicles. This could facilitate the subsequent scale-up preparation for clinical use. Herein, the protoplast-derived NVs are functionalized with pH-responsive PEG-conjugated phospholipid derivatives and TAM-targeting phospholipid derivatives, allowing for the precise delivery of Cas9-sgRNA RNP targeting Pik3cg (a critical modulator of macrophage phenotype) and CpG-rich genomic DNA fragments (natural TLR9 ligands) into TAMs. This dual targeting acts synergistically to redirect TAMs towards an anti-tumor M1-like phenotype. Importantly, these gene-edited TAMs using this bacterial vesicle-based delivery system are not susceptible to the hostile tumor microenvironment and exhibit a sustainable anti-tumor phenotype. Our one-step, self-assembly method for preparing a bacterial vesicle-based gene editing platform offers several advantages, including ease of preparation, high encapsulation efficiency, favorable safety and efficient targeted delivery. This platform may have great potential for developing cancer immunotherapy strategies.

## Results

### Preparation of *E. coli* protoplast-based NVs for TAM-specific *Pik3cg* genome editing

As shown in Fig. 1, plasmids encoding sg*Pik3cg* and *Cas9* were constructed and subsequently transformed into *E. coli*, in which sg*Pik3cg* is the sgRNA-1 (Supplementary Fig. 1). Following incubation with lysozyme, *E. coli* protoplasts loaded with the Cas9-sg*Pik3cg* complex were generated. Meanwhile, a pH-responsive phospholipid derivative of 1,2-distearoyl-sn-glycero-3-phosphorylethanolamine (DSPE)-hydrazone bond-PEG$_{2000}$ (DHP) and a TAM-targeted phospholipid derivative of DSPE-galactosamine (DGA) were synthesized and characterized (Supplementary Figs. 2 and 3). Rhodamine B-labeled DHP (RhB-DHP), Cy5.5-labeled DGA (Cy5.5-DGA) or fluorescein isothiocyanate-labeled DGA (FITC-DGA) were prepared (Supplementary Fig. 4). When protoplasts are physically extruded in the presence of amphiphilic molecules, such as DHP and DGA, their hydrophobic tails align with those of the disrupted protoplast membrane. Driven by non-covalent interactions, primarily hydrophobic force, the membrane lipid molecules and phospholipid derivatives (DHP and DGA) spontaneously organize into self-sealing bilayers in aqueous solutions. These bilayers effectively load Cas9-sg*Pik3cg* complexes, forming nanovesicles known as sg*Pik3cg*-DHP/DGA-NVs. The flow cytometry data indicated that the addition of 101.25 µmol DHP/DGA each into $1 \times 10^9$ CFU protoplasts reached the maximum decoration efficiency, which was used for the following generation of sg*Pik3cg*-DHP/DGA-NVs (Fig. 2a and Supplementary Fig. 5). Moreover, the results of fluorescence resonance energy transfer (FRET) experiments showed that both FITC-DGA and RhB-DHP were inserted into the membrane of NVs (Supplementary Fig. 6a). Differential scanning calorimetry (DSC) experiments observed that a phase transition temperature peak at 50.1 °C in sg*Pik3cg*-DHP/DGA-NVs but not in free NVs, which further confirmed the successful incorporation of DHP and DGA into NVs (Supplementary Fig. 6b). The morphology and diameter of protoplast-derived NVs were characterized by transmission electron microscopy (TEM), nanoparticle tracking analysis (NTA) and dynamic light scattering (DLS). Results in Fig. 2b, c and Supplementary Tables 1, 2 demonstrated that both free NVs and sg*Pik3cg*-DHP/DGA-NVs exhibited uniform nano-sized lipid-bilayered vesicular structures with a low polydispersity index (PDI) and average diameters of $152.20 \pm 2.93$ nm and $157.92 \pm 4.29$ nm (determined by DLS), respectively. These results indicated that incorporating DHP and DGA into free NVs caused a slight increase in diameter, without affecting dispersibility or uniformity. Furthermore, NVs subjected to eight days of incubation in PBS, enduring three freeze-thaw cycles, and incubation in serum-containing PBS for 24 h showed no significant changes in integrity and diameter size. This confirmed the outstanding stability of sg*Pik3cg*-DHP/DGA-NVs (Supplementary Fig. 7).

Next, we used flow cytometry and FRET experiments to investigate the pH-responsive degradation of DHP. The data in Fig. 2d, e and Supplementary Fig. 8 demonstrated a significant decrease in RhB-positive NVs at pH 6.5, attributed to the hydrazone linkage break of DHP in an acid environment and the removal of RhB-labeled PEG$_{2000}$ from NVs. The hydrazone linkage break of DHP after treated with pH 6.5 PBS was further confirmed by the reappearance of emission peak of FITC at 520 nm of NVs (Supplementary Fig. 8b). Moreover, to examine the stability of the hydrazone bond of DHP in 4T1 tumor tissue, we performed in situ injections of RhB-DHP/DGA-CFSE-labeled NVs into both tumor and adjacent tissue sites. Distinct localization of CFSE-labeled NVs (green) from the RhB-labeled PEG$_{2000}$ was observed in the acidic tumor tissue. However, the fluorescence signals of CFSE and RhB completely overlapped in the tumor-adjacent tissue due to its neutral pH environment (Fig. 2f). Crucially, these injections did not induce noticeable erythema, inflammatory cell infiltration, or upregulation of pro-inflammatory cytokines such as IL-6 and IL-1β (Supplementary Fig. 9), indicating that inflammation is not the primary factor responsible for PEG$_{2000}$ segments detachment from NVs. This also implies that sg*Pik3cg*-DHP/DGA-NVs exhibit favorable biocompatibility and safety.

In addition to these in situ vesicle injections, we conducted a comprehensive in vitro and in vivo safety evaluation of

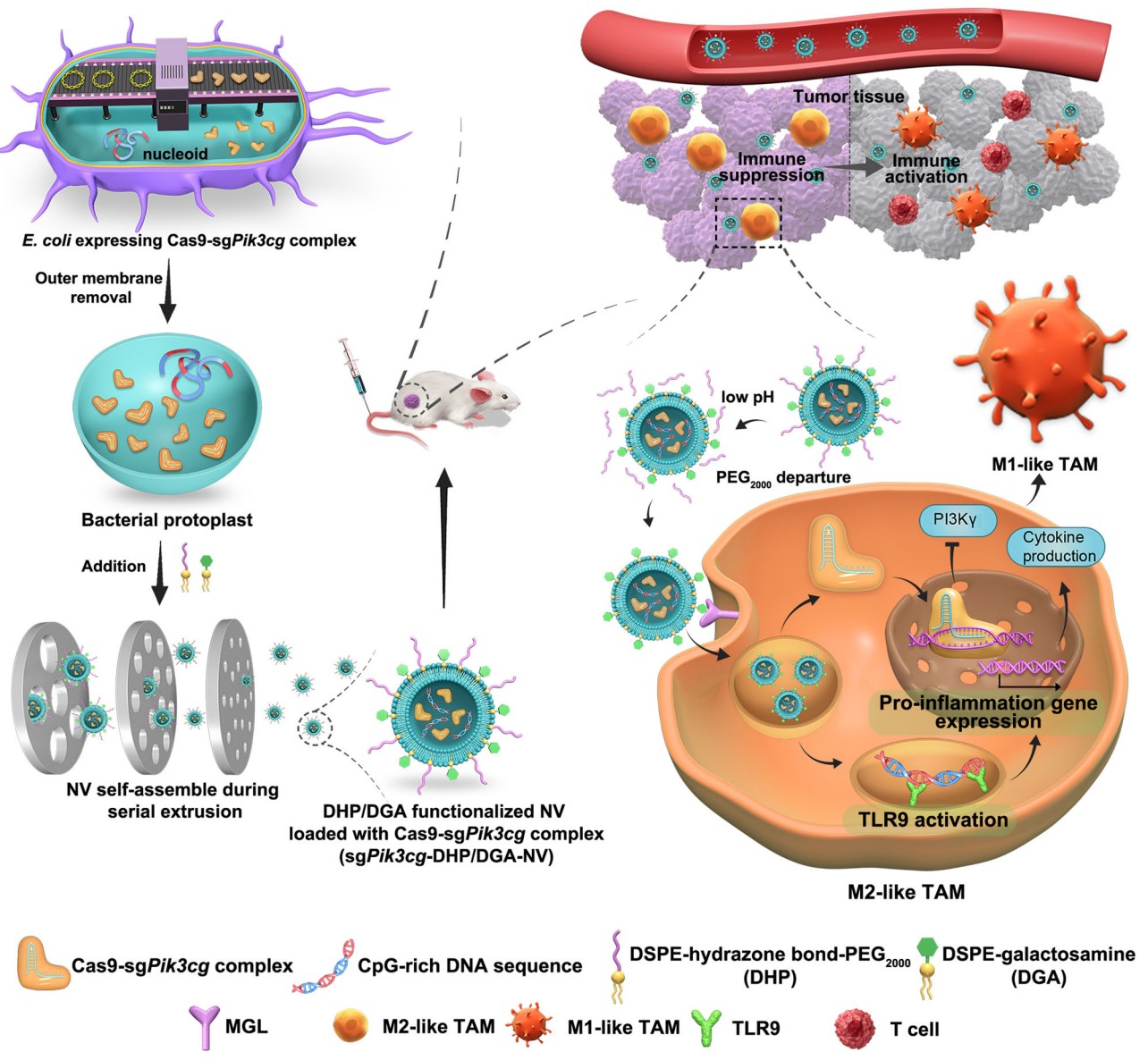

**Fig. 1 | Schematic design of *E. coli* protoplast-derived nanovesicles (sg*Pik3cg*-DHP/DGA-NVs) for TAM-selective genome editing to enhance anti-tumor efficacy.** In this illustration, we outline the creation of sg*Pik3cg*-DHP/DGA-NVs for targeted genome editing in tumor-associated macrophages (TAMs) to enhance anti-tumor effects. The process begins with the construction of *E. coli* expressing the Cas9-sg*Pik3cg* complex. Subsequently, the bacterial outer membrane, which possesses high endotoxicity, is removed. This results in the formation of sg*Pik3cg*-DHP/DGA-NVs, which encapsulate a substantial amount of Cas9-sg*Pik3cg* ribonucleoproteins (RNPs) and CpG-rich genomic DNA. These nanovesicles (NVs) are produced through a series of extrusion steps and are further modified with a pH-responsive phospholipid derivative (DHP) and a phospholipid derivative targeted specifically to TAM (DGA). Upon intravenous injection, sg*Pik3cg*-DHP/DGA-NVs accumulate in tumor tissues due to their prolonged circulation capability and the enhanced permeability and retention (EPR) effect. Within the acidic microenvironment of the tumor, $PEG_{2000}$ separates from DHP, triggering the recognition and internalization of DGA-functionalized NVs by TAMs via macrophage galactose-type lectin (MGL) receptor-mediated endocytosis. This process enables TAM-specific genome editing of *Pik3cg* and activation of toll-like receptor 9 (TLR9) in vivo, resulting in the reprogramming of M2-like TAMs into an anti-tumor M1-like phenotype and facilitating tumor immunotherapy.

sg*Pik3cg*-DHP/DGA-NVs, including analyses of LPS content, cytotoxicity, hemolytic toxicity, and evaluation of inflammatory factors in serum and immune cell populations in organs from mice following systemic administration of NVs. OMVs, naturally secreted by bacteria with known high toxicity, were utilized as controls in these experiments. ELISA assay indicated that NVs from bacterial protoplasts had lower LPS levels (Supplementary Fig. 10a). CCK8 assay showed that protoplast-derived NVs exhibited lower cytotoxicity compared to OMVs (Supplementary Fig. 10b). Given that the outer membrane of *E. coli* contains proteins with hemolytic toxicity, a hemolytic assay was conducted, which demonstrated that NVs from bacterial protoplasts induced minimal hemolysis (Supplementary Fig. 10c). Results in Fig. 2g and Supplementary Fig. 10e–g demonstrated that a single injection of sg*Pik3cg*-DHP/DGA-NVs led to an increase in serum cytokines (IL-6 and TNF-α) 2 h after intravenous NV injection. However, the cytokine levels and leukocyte ratio in blood, spleen and liver returned to normal within 24 h after NV administration. In contrast, mice treated with OMVs exhibited persistently elevated levels of inflammatory cytokines and neutrophil ratio in multiple organs. All mice were survived after intravenous NV injection, whereas mice treated with OMVs experienced fatalities (Supplementary Fig. 10d–g). These findings suggest that *E. coli* protoplast-derived NVs exhibit excellent biosafety.

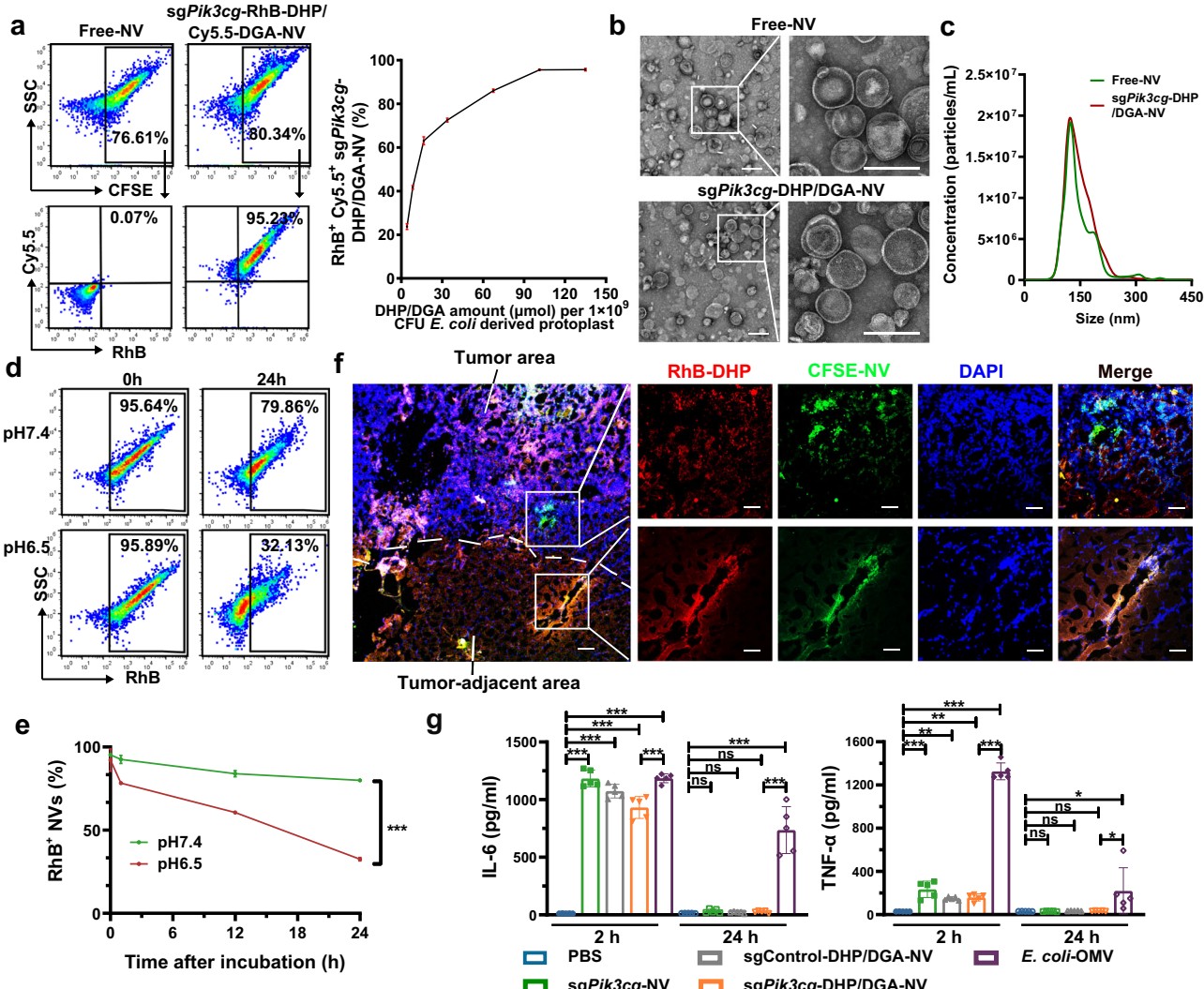

**Fig. 2 | The preparation and characterization of sg*Pik3cg*-DHP/DGA-NVs. a** A total of $1 \times 10^9$ colony forming unit (CFU) *E. coli* derived protoplasts were added with indicated amounts of RhB-DHP and Cy5.5-DGA for physical extrusion. The decoration efficiency and quantitative analysis of RhB-DHP and Cy5.5-DGA for nanovesicles were determined through flow cytometry analysis. n = 3 biologically independent samples. **b** Representative TEM images illustrate *sgPik3cg*-NVs, both with and without DHP/DGA decoration. Scale bar, 200 nm. **c** The diameters of sg*Pik3cg*-NVs and sg*Pik3cg*-DHP/DGA-NVs were analyzed by Nanosight. For (**b**, **c**), experiments were independently conducted three times with similar results. **d** The ratio of RhB-positive NVs was quantified through flow cytometry analysis after treating $1 \times 10^{10}$ CFSE-labeled NVs with RhB-DHP and DGA in 100 μL of pH 6.5 PBS at indicated time points (0 h and 24 h). **e** Quantitative analysis of sg*Pik3cg*-RhB-DHP/Cy5.5-DGA-NVs treated with different pH values of PBS at indicated time

points. n = 3 biologically independent samples for (**d**, **e**). Statistical analysis was performed using two-way ANOVA with Bonferroni's multiple comparison test. **f** Direct injection of 50 μL PBS containing $5 \times 10^9$ CFSE-labeled sg*Pik3cg*-RhB-DHP/DGA-NVs into 4T1 tumor tissues and tumor-adjacent tissues. Corresponding samples were harvested from tumor-bearing mice at 24 h post-injection for fluorescence photography. The representative images presented are from a sample size of n = 3 mice. Scale bar = 50 μm. **g** Measurement of IL-6 and TNF-α levels in 100 μL serum of 4T1 tumor-bearing mice at 2 h and 24 h after vein injections of 100 μL PBS containing different types of NVs (dose: $1 \times 10^{10}$ per mouse). n = 5 mice per group. Statistical analysis was performed using one-way ANOVA with Dunnett's multiple comparison test. Data are represented as mean ± SD. *$P < 0.05$, **$P < 0.01$ and ***$P < 0.001$, ns, no significant change. The exact *P*-value and source data are provided as a Source Data file.

## Analysis of the components of *E. coli* protoplast-derived NVs

The nucleotide content of NVs was analyzed by electrophoresis technique and PCR assay. Electrophoresis results in Fig. 3a, b and Supplementary Fig. 11a, b revealed that both DNA and RNA in protoplast-derived NVs had a broader but smaller size distribution compared to *E. coli*, suggesting that the extrusion process led to nucleic acid molecule breakage. The presence of CpG-rich genomic DNA in NVs was predicted bioinformatically (Supplementary Data 1) and confirmed by PCR assay[17] (Fig. 3c). The DNA derived from NVs, including CpG-rich genomic DNA sequences, could upregulate TNF-α levels in bone marrow-derived macrophages (BMDMs). Simultaneously, the inhibitor of TLR9 (ODN2088) significantly reduced the TNF-α levels induced by NVs-encapsulated DNA, suggesting that the immunostimulatory effect

of NVs-encapsulated DNA, containing CpG-rich genomic DNA, depended on TLR9 (Supplementary Fig. 12). Moreover, PCR analysis confirmed the presence of sgRNA for *Pik3cg* in both *E. coli* and NVs (Fig. 3d). We used protein liquid chromatography-mass spectrometry (LC-MS) combined with label free-quantification (LFQ) and western blotting to identify the protein contents. We applied intensity-based absolute quantification (iBAQ) to rank the abundance of distinct proteins within each group. The top 200 abundant proteins were listed in Supplementary Data 2–5. Additionally, the subcellular localization of the top 200 abundant proteins in each sample was analyzed. Compared to the types found in OMVs, the sg*Pik3cg*-DHP/DGA-NVs contained fewer kinds of outer membrane and periplasmic proteins (Fig. 3e). Furthermore, we compared the outer membrane proteins

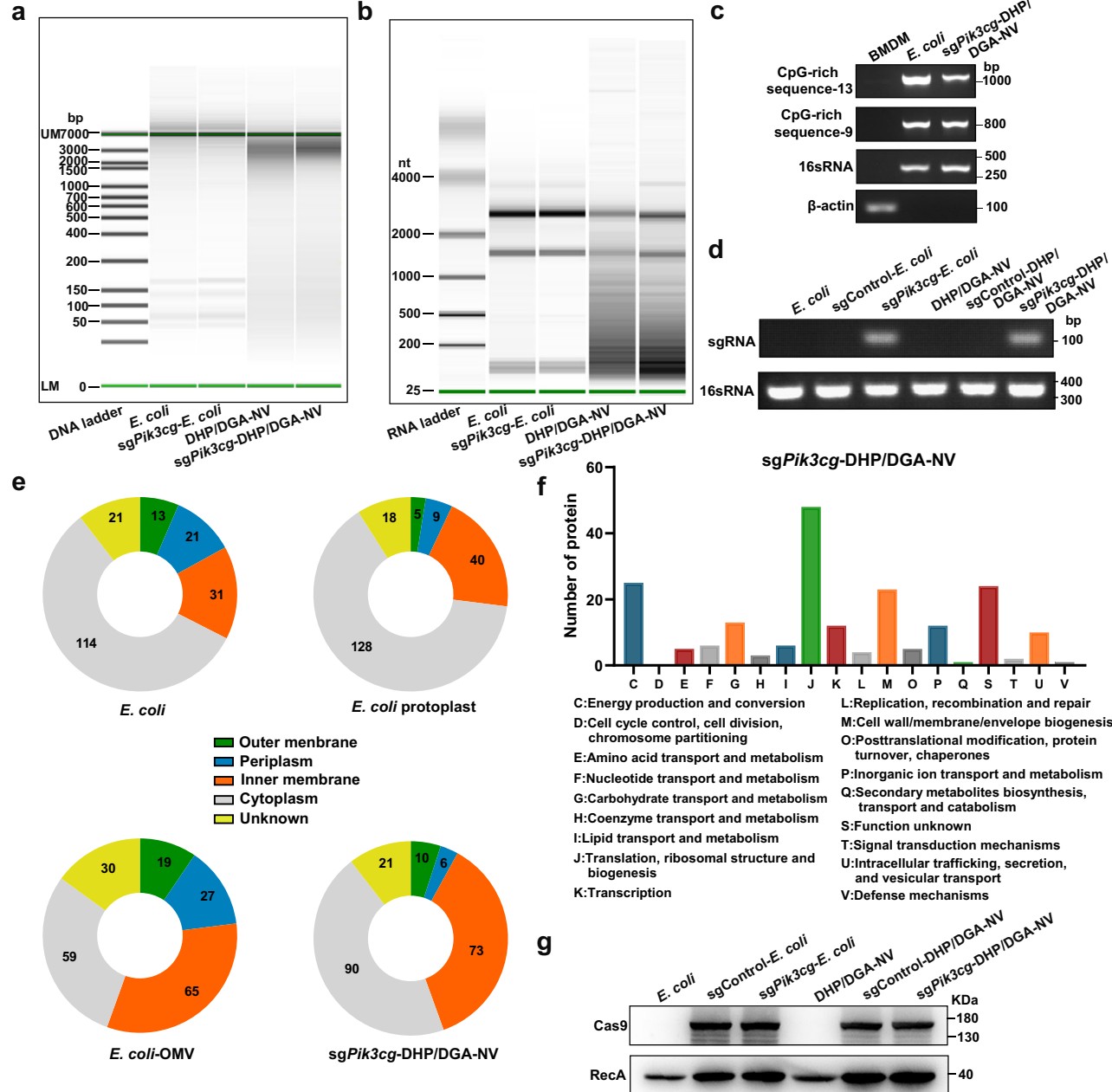

**Fig. 3 | The components analysis of sg*Pik3cg*-DHP/DGA-NVs. a, b** Size distribution and abundance of DNA and RNA from *E. coli*, protoplast, OMVs and sg*Pik3cg*-DHP/DGA-NVs were determined using LabChip analyzer and Agilent 2100 bioanalyzer. **c, d** The presence of CpG-rich genomic DNA sequences and sgRNA targeting *Pik3cg* was confirmed through PCR amplification and agarose gel electrophoresis. **e** Subcellular localization of the top 200 abundant proteins, ranked according to their iBAQ protein intensity, was identified in *E. coli*, protoplasts, OMVs and sg*Pik3cg*-DHP/DGA-NVs through proteomic analysis, with 100 μg of protein used for each sample. A pie chart illustrates the variety of protein types categorized by their subcellular localization in each sample. **f** Analysis of the Clusters of Orthologous Groups (COG) was conducted for the top 200 abundant proteins identified in sg*Pik3cg*-DHP/DGA-NVs. **g** The presence of Cas9 protein in *E. coli* and sg*Pik3cg*-DHP/DGA-NVs was determined using western blotting. All experiments for (**a–g**) were independently repeated three times, yielding consistent results. Source data are provided as a Source Data file.

and periplasmic proteins between OMVs and sg*Pik3cg*-DHP/DGA-NVs using LFQ intensity (Supplementary Fig. 13). The results indicated a significant reduction in the abundance of specific outer membrane and periplasm proteins, such as OmpA and DcrB, in protoplast-derived NVs compared to *E. coli*-derived OMVs, based on the log2 transformed LFQ intensities of these proteins. Based on functional classification using the Clusters of Orthologous Groups of proteins (COG), most identified proteins in sg*Pik3cg*-DHP/DGA-NVs were associated with cellular processes like metabolism, transporter activity, and translation (Fig. 3f).

Western blotting results confirmed the presence of Cas9 protein in both *E. coli* and NVs (Fig. 3g).

Additionally, we calculated the contents of Cas9 and sg*Pik3cg* in *E. coli*, protoplasts derived NVs and OMVs (Supplementary Table 3). The packaging efficiencies of Cas9 and sg*Pik3cg* in sg*Pik3cg*-DHP/DGA-NVs are 23.00% and 24.46%, respectively, whereas their packaging efficiencies within OMVs are approximately 0.87% and 0.54% (Supplementary Fig. 14a). Cas9 protein and sg*Pik3cg* content in sg*Pik3cg*-DHP/DGA-NVs from $1 \times 10^9$ CFU bacteria are about 26 times higher than

those in OMVs (Supplementary Fig. 14b). Moreover, $1 \times 10^9$ CFU bacteria can produce sg*Pik3cg*-DHP/DGA-NVs with a total protein amount of 117.5 μg, but can only yield 5.42 μg of OMVs in protein amount after 24 h of cultivation. In summary, these findings highlight the superior performance of vesicles from protoplasts in large-scale preparation.

## In vitro targeted delivery of sg*Pik3cg*-DHP/DGA-NVs to M2-like macrophages

CFSE-labeled DGA-NVs were used to monitor NVs' efficiency in entering M2-like BMDMs (M2-BMDMs). Flow cytometry analysis and microscopical images demonstrated that DGA decoration greatly enhanced NVs' uptake by macrophages (Fig. 4a–d). DGA utilized as a ligand to direct the NVs toward TAMs that highly express macrophage galactose-type lectin (MGL)[18–20]. The entry efficiency of sg*Pik3cg*-DHP/DGA-NVs into specific cell types correlated positively with their MGL expression levels (Supplementary Figs. 15, 16). The presence of N-acetylgalactosamine (GalNAc), but not N-acetylglucosamine (GlcNAc), and interference with MGL expression led to decreased entry efficiency of NVs into M2-BMDMs (Fig. 4a–d and Supplementary Fig. 17). Additionally, the uptake efficiency of sg*Pik3cg*-NVs into M2-BMDMs was about 23.68%. This may be attributed to the various internalization mechanisms such as membrane fusion, macropinocytosis or lipid-raft-mediated uptake[21]. The modification of DHP enhanced NVs' hydrophilicity but hindered their internalization by macrophages at pH 7.4. However, incubation in acidic PBS (pH 6.5) caused $PEG_{2000}$ fragment to detach from NVs, exposing DGA. The cellular efficiency of sg*Pik3cg*-DHP/DGA-NVs pre-treated with pH 6.5 PBS recovered to 60.16%, similar to that of the sg*Pik3cg*-DGA-NV group (Fig. 4e, f). Fluorescence images and intensity detection yielded consistent results (Fig. 4g, h).

sg*Pik3cg*-DHP/DGA-NVs pre-treated with pH 6.5 PBS were used in the following experiments. To determine the subcellular localization, lysosomal NVs were examined using fluorescence co-localization assays. Supplementary Fig. 18 exhibited poor correlation between lysosomal dye lysotracker and CFSE-labeled NVs at 12 h after NVs incubation, indicating successful lysosomal escape. Immunofluorescence co-staining revealed green fluorescence-labeled Cas9 overlapping with the macrophage nucleus at 12 h post NVs treatment, demonstrating Cas9 entry mediated by nuclear localization sequence (NLS) (Fig. 4i). Moreover, the results of western blotting and PCR indicated the detection of both Cas9 protein and sgRNA sequence in total cells and nucleus components (Fig. 4j, k). Additionally, PCR amplification and agarose gel electrophoresis demonstrated the transfer of CpG-rich genomic DNA fragments into macrophages by sg*Pik3cg*-DHP/DGA-NVs pre-treated with pH 6.5 PBS (Fig. 4l).

## In vitro repolarization of M2-like macrophages with sg*Pik3cg*-DHP/DGA-NVs

In vitro, sg*Pik3cg*-DHP/DGA-NVs mediated *Pik3cg* genome editing was assessed by T7 Endonuclease I (T7E1) assay and next-generation sequencing (NGS). Figure 5a showed evident *Pik3cg* cleavage after T7E1 digestion, with an indel rate of approximately 28.5%. Sanger sequencing and NGS of PCR amplicons at the editing site are summarized in Supplementary Fig. 19a–d. T7E1 assay on potential off-target sites showed negligible off-target effects of sg*Pik3cg* (Supplementary Fig. 19e). Western blotting analysis further demonstrated the downregulation of PI3Kγ protein following treatment with NVs containing sg*Pik3cg* compared to sgControl-containing NVs (Fig. 5b). In addition, an NF-κB luciferase reporter assay was applied to evaluate whether NVs' immunostimulatory effects from CpG rich-DNA fragments depended on TLR9. NVs containing sgControl activated luciferase activity in TLR9-overexpressed 293 T cells. However, the stimulating effect of NVs was partly abolished by the addition of TLR9 inhibitor (ODN2088) (Fig. 5c). We performed western blotting analysis to examine PI3Kγ and TLR9 signaling pathway-related markers, including

p-AKT, p-C/EBPβ and p-p65 (PI3Kγ), p-IRAK4 (TLR9), p-TAK1 (TNF), p-STAT1 (IFN) and p-STAT3 (IL-6) (Fig. 5d and Supplementary Fig. 20). Gene editing of *Pik3cg* by sg*Pik3cg*-DHP/DGA-NVs significantly reduced p-AKT and p-C/EBPβ levels and increased p-p65 levels while activating the TLR9 pathway, characterized by elevated p-IRAK4 in the sg*Pik3cg*-DHP/DGA-NVs-treated group (Fig. 5d). Furthermore, the conversion efficiency of PI3Kγ substrate PIP2 to PIP3 was markedly reduced in the sg*Pik3cg*-DHP/DGA-NVs treated group, consistent with the western blotting results (Fig. 5e). Consequently, sg*Pik3cg*-DHP/DGA-NVs stimulation reversed the M2-BMDMs phenotype, upregulating M1 markers (CD86 and iNOS), immunostimulatory cytokines (TNF-α, IL-12, IFN-γ and IL-6) and downstream signaling molecules (p-TAK1, p-STAT1, and p-STAT3). Conversely, M2 marker (CD206 and Arg1) and immunoregulatory cytokines (IL-10 and TGF-β1) in M2-BMDMs were reduced after sg*Pik3cg*-DHP/DGA-NVs treatment (Fig. 5d–h and Supplementary Fig. 20). To determine whether the repolarization of macrophage phenotype by NVs depends on suppressing PI3Kγ expression and activating TLR9 signaling pathway, we used ODN1826 (TLR9 agonist) and IPI549 (PI3Kγ inhibitor). Treatment with either ODN1826 or IPI549 showed lower stimulation compared to NVs containing sg*Pik3cg*. Additionally, the addition of TLR9 inhibitor (ODN2088) partly abolished the activating effects of NVs containing sg*Pik3cg* (Fig. 5d–h).

To assess whether *Pik3cg* gene-edited macrophages could maintain their immunostimulatory phenotype in an immunosuppressive microenvironment, we utilized 4T1 tumor cell supernatant rich in immunoregulatory cytokines (TGF-β, IL-33, IL-4 and IL-10) to mimic the in vivo tumor setting (Supplementary Fig. 21a)[22]. After treating macrophages with NVs, we incubated them with 4T1 tumor cell supernatant for 24 h, and then analyzed their phenotype. sg*Pik3cg*-DHP/DGA-NVs treated macrophages still maintained immune-stimulatory phenotype, characterized by high levels of M1 markers and proinflammatory cytokines, and low levels of M2 markers and immunoregulatory cytokines. However, the activating phenotype of IPI549-treated macrophages was impaired after exposure to tumor cells supernatant, indicating the persistent ability of the gene editing system to sustain the M1-like phenotype of macrophages in an immunosuppressive environment (Supplementary Fig. 21b, c).

## In vivo delivery of sg*Pik3cg*-DHP/DGA-NVs to TAMs activated M1-like phenotype

To examine NVs' biodistribution, fluorescence images of mice and corresponding organs were captured at indicated time points after intravenous injection of Cy5 labeled NVs. Figure 6a, b showed substantial accumulation of sg*Pik3cg*-DHP/DGA-NVs within the tumor site, with modest vesicle accumulation in the sg*Pik3cg*-NV and sg*Pik3cg*-DGA-NV treated groups. Pharmacokinetics study demonstrated the longer circulation time of sg*Pik3cg*-DHP/DGA-NVs compared to sg*Pik3cg*-NVs and sg*Pik3cg*-DGA-NVs (Fig. 6c). To further quantify vesicle bio-distribution, we collected and homogenized tissues from mice injected with Cy5 labeled NVs and subjected them to fluorescence quantification analysis (Fig. 6d and Supplementary Fig. 22a). These results corroborated previous in vivo imaging and organ photographs, confirming the substantial presence of NVs within the tumor tissue in the sg*Pik3cg*-DHP/DGA-NVs treatment group. Results in Fig. 6e, f confirmed that most sg*Pik3cg*-DHP/DGA-NVs accumulated in the F4/80+ cells, representing TAMs. Consistent with fluorescent images, flow cytometry analysis of tumor leukocytes revealed that a single administration of sg*Pik3cg*-DHP/DGA-NVs resulted in a peak of Cy5-positive TAMs at 72 h, with 6.62% of macrophages remaining Cy5 positivity after 10 days of treatment. In contrast, fewer TAMs were Cy5 positive in mice injected with decoration-free sg*Pik3cg*-NVs or sg*Pik3cg*-DGA-NVs (Fig. 6g–j). Clear T7E1 cleavage bands were detected in TAM DNA from mice treated with sg*Pik3cg*-DHP/DGA-NVs.

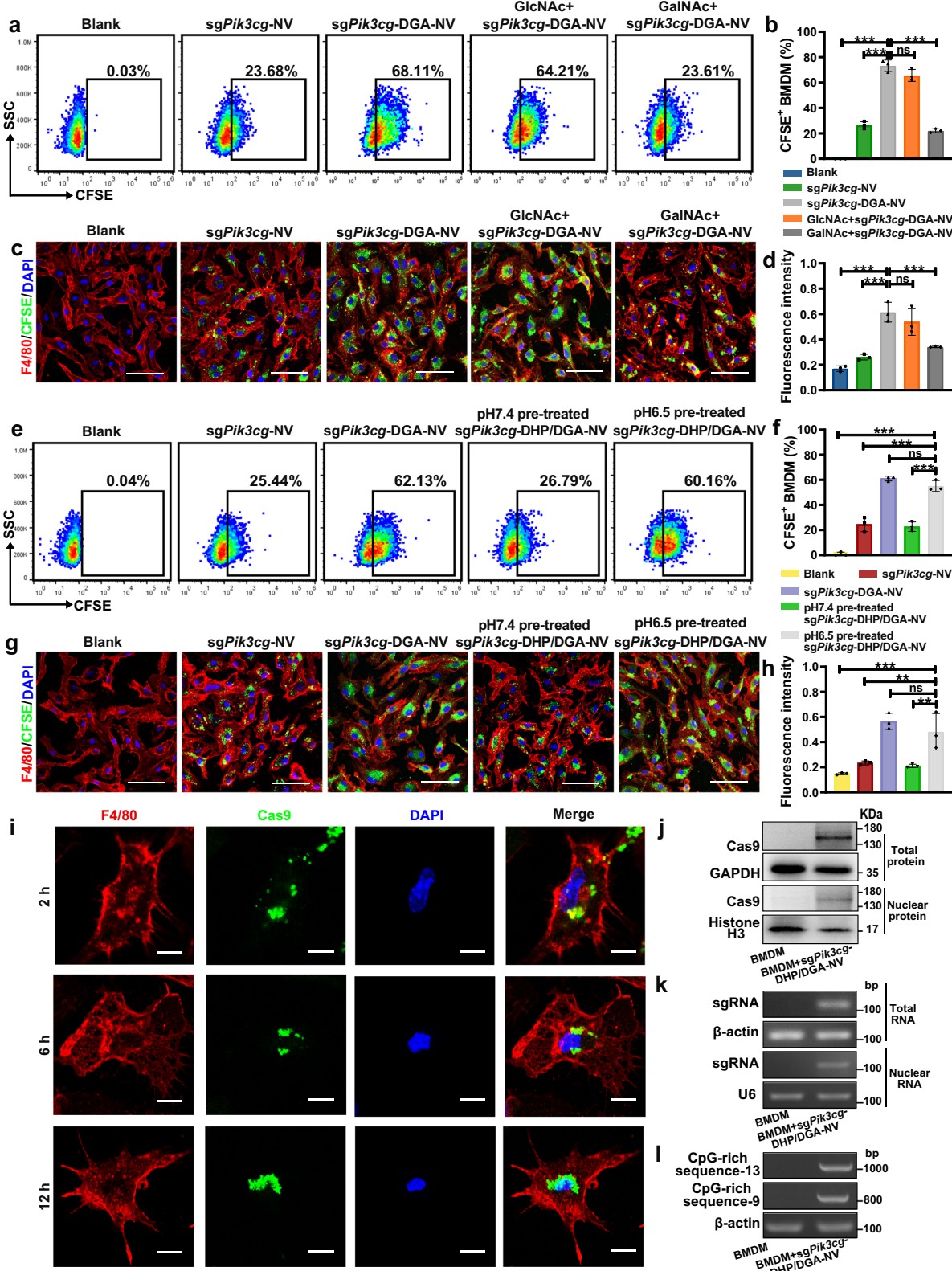

Similarly, the corresponding indel rate in TAMs peaked at 15.3% after 72 h and remained at approximately 2.5% after 10 days (Fig. 6k, l). NGS also confirmed successful *Pik3cg* gene-editing in TAMs by sg*Pik3cg*-DHP/DGA-NVs, with the representative mutation pattern shown in Supplementary Fig. 23. Flow cytometry and T7E1 assay were also applied to detect Cy5-positive and gene edited cells in tumor and other organs, including peripheral blood, liver, and spleen. Supplementary

Fig. 22b, c showed the highest Cy5-positive cell proportions in hepatocytes, liver monocytes and endothelial cells, comprising approximately 14% of the total, with an editing efficiency approaching 3%. At 72 h, aside from tumor cells, hepatocytes, and hepatic endothelial cells, editing proportions in other organs became negligible. Repeated administration of Cy5⁺ sg*Pik3cg*-DHP/DGA-NVs led to fluctuating fluorescence levels in plasma and a progressive rise in tumor

**Fig. 4 | In vitro uptake of NVs by M2-like macrophages via MGL-mediated endocytosis. a–c** $5 \times 10^5$ M2-like bone marrow-derived macrophages (M2-BMDMs) were seeded in 24-well plates and treated with $6 \times 10^8$ CFSE-sg*Pik3cg*-DGA-NVs for 3 h, followed by flow cytometry analysis and fluorescent microscopy. In some cases, macrophages were pre-incubated with GlcNAc or GalNAc (100 mmol/L) for 1 h before NVs treatment. Red, F4/80; green, CFSE labeled NVs; blue, DAPI nuclear staining. Scaled bar = 50 μm. *n* = 3 biologically independent samples. **d** A total of $1 \times 10^5$ macrophages with above mentioned treatments was harvested, lysed and the fluorescence intensity was quantified using a microplate reader (Em: 488 nm; Ex: 530 nm). *n* = 3 biologically independent samples. **e–h** $5 \times 10^5$ M2-BMDMs were treated with $6 \times 10^8$ CFSE-sg*Pik3cg*-DGA-NVs or CFSE-sg*Pik3cg*-DHP/DGA-NVs (pre-treated with pH 6.5 or 7.4 PBS for 24 h) for 3 h and then subjected to flow cytometry analysis, microphotography and fluorescence intensity quantification. Red, F4/80; green, CFSE labeled NVs; blue, DAPI nuclear staining. Scaled bar = 50 μm. *n* = 3 biologically independent samples. **i** M2-BMDMs subjected to the aforementioned NVs treatment were fixed, stained with fluorescent-labeled F4/80 and Cas9 antibodies and imaged at indicated time points. Red, F4/80; green, Cas9; blue, DAPI nuclear staining. Scaled bar = 10 μm. **j, k** The levels of Cas9 and sgRNA targeting *Pik3cg* in $5 \times 10^5$ M2-BMDMs treated with $6 \times 10^8$ sg*Pik3cg*-DHP/DGA-NVs (pH 6.5 PBS pre-treatment) for 3 h after which the medium was replaced with fresh medium for another 9 h. The cells were examined by western blotting and agarose gel electrophoresis. **l** The NVs-mediated delivery of CpG-rich genomic DNA fragments into $5 \times 10^5$ M2-BMDMs incubated with $6 \times 10^8$ NVs for 3 h was detected by PCR amplification and agarose gel electrophoresis. The experiments for (**i**–**l**) were independently repeated three times with similar results. Data are presented as the means ± SD. Statistical analyses were performed using one-way ANOVA with Dunnett's multiple comparison test. *$P < 0.05$, **$P < 0.01$ and ***$P < 0.001$, ns, no significant change. The exact *P*-value and source data are provided as a Source Data file.

fluorescence (Supplementary Fig. 24a–c). The proportion of Cy5+ TAMs reached 73.64%, with a corresponding TAM editing efficiency of 34.4% (Supplementary Fig. 24d–g). These findings indicate precise editing of TAMs by sg*Pik3cg*-DHP/DGA-NVs.

### sg*Pik3cg*-DHP/DGA-NVs treatment inhibited tumor growth

The therapeutic effects of sg*Pik3cg*-DHP/DGA-NVs were initially evaluated in 4T1 breast cancer mouse models. Intravenous injection of sg*Pik3cg*-DHP/DGA-NVs reduced tumor weight and size without noticeable side effects (Fig. 7a–d and Supplementary Fig. 25). sg*Pik3cg*-DHP/DGA-NVs demonstrated superior therapeutic effectiveness compared to IPI549 treatment alone. Interestingly, combining sgControl-DHP/DGA-NVs with IPI549 yielded similar therapeutic outcomes to sg*Pik3cg*-DHP/DGA-NVs treatment alone. H&E staining of tumor sections indicated that sg*Pik3cg*-DHP/DGA-NVs treatment induced the largest necrotic area within tumors (Fig. 7e). Enzyme activity assay and western blotting confirmed the suppression of PI3Kγ-related downstream signaling pathways and the activation of TLR9 signaling pathways in TAMs from mice treated with sg*Pik3cg*-DHP/DGA-NVs (Fig. 7f, g). Furthermore, results in Fig. 7h–j and Supplementary Fig. 26 demonstrated that sg*Pik3cg*-DHP/DGA-NVs treatment shifted TAMs toward an immunostimulatory M1-like phenotype, characterized by increased M1 markers, immunostimulatory cytokines and activated IFN-γ and TNF-α signaling pathways, along with reduced M2 markers and immunoregulatory cytokines. In comparison, IPI549 treatment showed less stimulatory effects on TAMs (Fig. 7g–j and Supplementary Fig. 26). Both TLR9 gene deletion in mice and the application of a TLR9 antagonist (ODN2088) impaired the therapeutic effects of sg*Pik3cg*-DHP/DGA-NVs, resulting in therapeutic effects similar to those of IPI549 (Supplementary Figs. 27, 28). These results emphasize the pivotal roles of TLR9 pathway activation and PI3Kγ pathway blockade in the therapeutic efficacy of sg*Pik3cg*-DHP/DGA-NVs.

### sg*Pik3cg*-DHP/DGA-NVs therapy reshaped the tumor microenvironment

As TAM plays an important role in shaping the tumor microenvironment by expressing high levels of immunosuppressive cytokines and tumor-promoting factors[23,24], we assessed the impact of sg*Pik3cg*-DHP/DGA-NVs on the 4T1 tumor microenvironment through transcriptome sequencing. This analysis revealed a multitude of genes involved in immune response and leukocyte activation, including pathways related to TNF-α, IFN-γ, TLR9, NF-κB activation, and T cell activation (Fig. 8a, b and Supplementary Data 6–8). Further validation through GSEA analysis solidified the positive association between these pathways and the effects of sg*Pik3cg*-DHP/DGA-NVs treatment (Fig. 8c, Supplementary Fig. 29 and Supplementary Data 9).

ELISA and immunopathological staining of 4T1 tumor tissue treated with sg*Pik3cg*-DHP/DGA-NVs showed a significant increase in immunostimulatory cytokines and a decrease in immunoregulatory cytokines (Fig. 8d and Supplementary Fig. 30a). Flow cytometry data indicated a substantial increase in the ratio of CD4+ and CD8+ T cells in both the tumor and peripheral blood, along with heightened expression of activation and proliferation markers (IFN-γ+, Ki67+, and Granzyme+), particularly within 4T1 tumor tissues following sg*Pik3cg*-DHP/DGA-NVs treatment (Fig. 8e and Supplementary Figs. 30b and 31). Additionally, there was a significant increase in the proportion of CD80+ and CD86+ dendritic cells (DCs) within mouse tumor tissues after sg*Pik3cg*-DHP/DGA-NVs treatment, indicating DC activation (Supplementary Fig. 30c). These cytokines and immune cell activation patterns were consistent in sg*Pik3cg*-DHP/DGA-NVs treated MC38 tumor tissue (Supplementary Fig. 32). These findings suggest that the reversal of TAM phenotype facilitated by sg*Pik3cg*-DHP/DGA-NVs leads to a transformation of the entire tumor microenvironment from a "cold" state to a "hot" state. Given the cytotoxic impact of heightened TNF-α levels and massive T cell infiltration on tumor cells within a short timeframe, we employed neutralizing antibodies targeting TNF-α and CD8. This resulted in a reduction in the therapeutic efficacy of sg*Pik3cg*-DHP/DGA-NVs, indicating that the elevation of immunostimulatory cytokines and activated T cell infiltration mediated by NVs is crucial for its anticancer activities (Supplementary Figs. 33, 34). Furthermore, the co-administration of a PD-1 inhibitor and sg*Pik3cg*-DHP/DGA-NVs demonstrated superior efficacy compared to using PD-1 inhibitor or sg*Pik3cg*-DHP/DGA-NVs alone (Supplementary Fig. 35). These findings suggest that the combination of PI3Kγ blockade with PD-1 inhibitors holds significant promise for cancer immunotherapy.

## Discussion

Although autologous cell-based gene editing therapy is advancing rapidly in clinical trials, CRISPR-Cas9 system for clinical tumor treatment is still in the early stages of development[25]. In order to achieve effective and precise cancer treatment, the CRISPR-Cas9 components must penetrate different physical barriers to reach the target cells, and the Cas9 protein and sgRNA must be transported into the nucleus for gene editing. Therefore, a safe and efficient delivery system is crucial for the success of CRISPR-Cas9-mediated editing therapeutics. Although non-viral carriers have been rapidly developed and show great potential for CRISPR-Cas9 delivery, further improvement are urgently required to enhance their in vivo targeting ability. Moreover, the big size and charged surface of CRISPR-Cas9 components make it difficult to condense into a nano-sized delivery carrier. Biological vesicles might be a more promising delivery carrier for CRISPR-Cas9 therapy[11]. Several approaches have utilized mammalian cell-derived vesicles (exosomes, etc.) as the carrier of gene editing system[26]. However, mammalian cell-derived vesicles have been shown to be involved in a variety of physiological responses, and their function complexity restricts their application in drug delivery development[27].

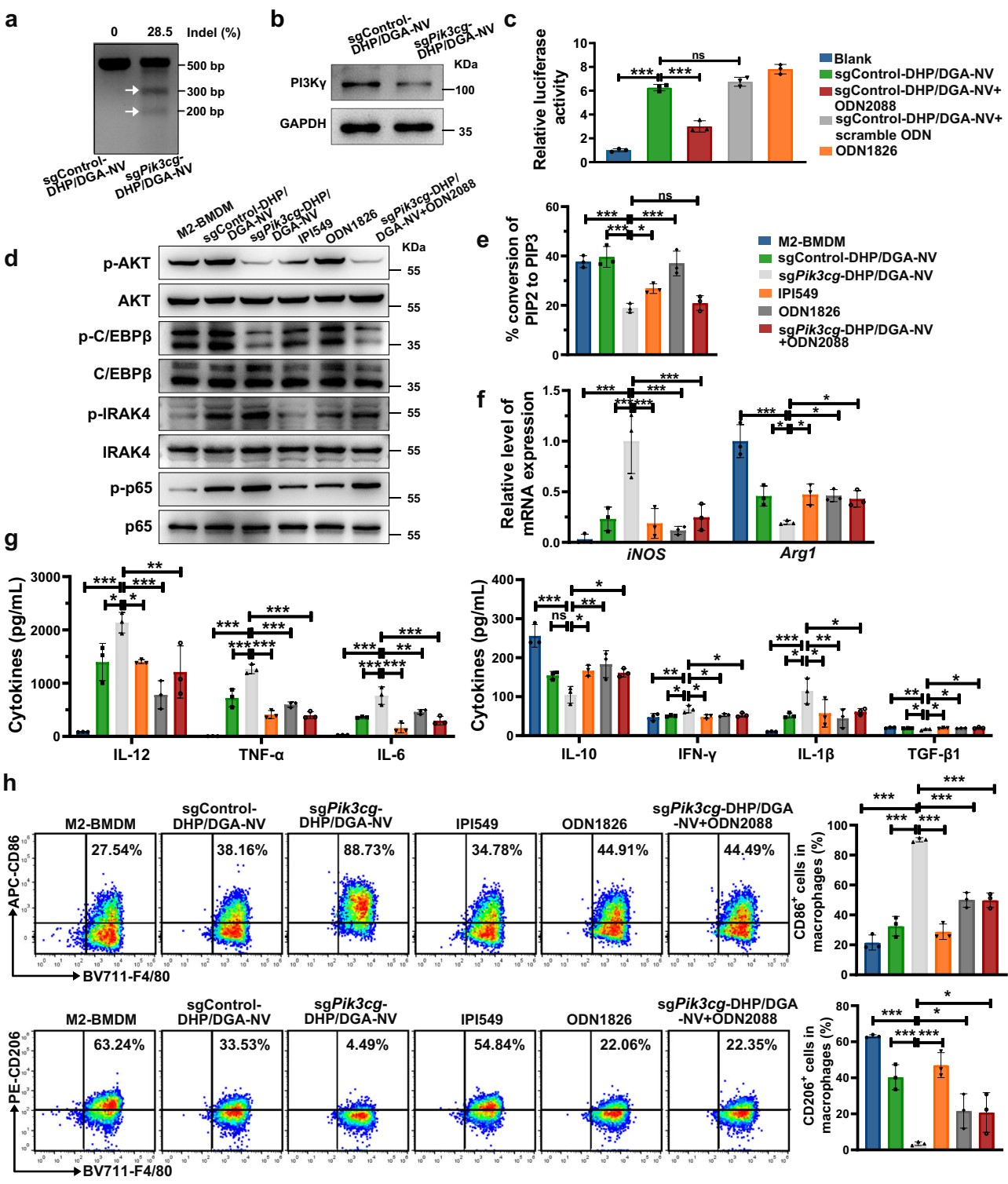

Additionally, the low production yield of naturally secreted extracellular vesicles results in a high cost for their preparation[28]. However, lytic bacterial-derived vesicles could potentially offer a promising alternative. Their parent cells can be efficiently produced in large quantities through fermentation, and the yield of vesicles obtained through bacterial lysis is significantly higher compared to those obtained through natural secretion. In this study, we used engineered bacteria as the site for both gene editing tool production and carrier materials construction, which further simplified the preparation process. Through physical extrusion, the protoplast derived NVs showed a highly uniform morphology of spherical bilayers. The yield of

protoplast-derived vesicles (117.5 μg of sg*Pik3cg*-DHP/DGA-NVs in protein amount from $1 \times 10^9$ CFU bacteria) significantly exceeds that of naturally-produced bacterial vesicles (5.42 μg of OMVs in protein amount from $1 \times 10^9$ CFU bacteria), according to our study and previous research[15]. Unlike the selective packaging seen into OMVs (packaging efficiency: 0.87% for Cas9 and 0.54% for sg*Pik3cg*), the physical extrusion process could ensure the effective loading of Cas9-sg*Pik3cg* RNP (packaging efficiency: 23.00% for Cas9 and 24.46% for sg*Pik3cg*) and bacterial CpG-rich DNA fragments. This high-efficiency loading occurs without affecting the biological activities of the components, as verified by our analyses. Physical extrusion, therefore,

**Fig. 5 | The genome editing efficiency of *Pik3cg* and phenotypic analysis in M2-like macrophages after sg*Pik3cg*-DHP/DGA-NVs treatment. a, b** $1.5 \times 10^6$ M2-BMDMs in 6-well plates were treated with $1.8 \times 10^9$ sg*Pik3cg*-DHP/DGA-NVs (pH 6.5 PBS pre-treatment) for 6 h, followed by replacement with fresh medium. Macrophages were harvested 48 h post-incubation with NVs for T7E1 analysis to assess indel formation and western blotting to determine PI3Kγ levels. The experiments were repeated three times independently with similar results. **c** $5 \times 10^5$ 293 T cells transfected by TLR9 overexpression plasmid, NF-κB reporter plasmid and β-gal reference plasmid (each plasmid: 2 μg) for 24 h were further treated with $1.8 \times 10^9$ different types of NVs or ODN1826 (TLR9 agonist, 10 μmol/L) for 6 h, followed by replacement with fresh medium for another 18 h to assess luciferase activity. In some cases, ODN2088 (TLR9 inhibitor, 10 μmol/L) and the corresponding scramble control were added 30 min before NVs treatment and co-incubated for another 24 h. $n = 3$ biologically independent samples. **d** $1.5 \times 10^6$ M2-BMDMs were treated with $1.8 \times 10^9$ different types of NVs for 6 h, followed by replacement with fresh medium for another 42 h. ODN1826 (10 μmol/L) or IPI549 (PI3Kγ inhibitor, 1 μmol/L) were added for 48 h, and then cells were harvested for western blotting to examine the downstream proteins of PI3Kγ and TLR9 (except p-IRAK4). p-IRAK4/IRAK4 was examined 1 h after NVs or reagents treatment. In some cases, 10 μmol/L ODN2088 was added 30 min before NVs treatment and co-incubated for 48 h. The experiments were repeated three times independently with similar results. **e–h** M2-BMDMs with the above-mentioned NVs and IPI549 treatments were used to examine PIP2 transition ratio, macrophage phenotype markers by RT-qPCR and flow cytometry, and cytokine levels by ELISA. $n = 3$ biologically independent samples. Data are represented as mean ± SD. Statistical analyses were performed using one-way ANOVA with Dunnett's multiple comparison test. *$P < 0.05$, **$P < 0.01$ and ***$P < 0.001$, ns, no significant change. The exact $P$-value and source data are provided as a Source Data file.

represents a method for the rupture and subsequently self-assemble into NVs. Overall, the ease of production, high yield and efficient loading make protoplast-derived vesicles a promising avenue for large-scale clinical applications.

Bio-safety is the primary concern when utilizing bacterial-derived vesicles for therapeutic applications, such as, OMVs, which are natural extracellular vesicles derived from the outer membrane containing most virulent factors[29]. Systemic injection of OMVs could induce sepsis-like symptoms, including cytokines storm, which may be particularly dangerous for tumor patients who are already in poor physical condition[30]. Although genetic engineering of genes involved in LPS biosynthesis could reduce OMVs toxicity, other virulence factors, such as bacterial adhesins, proteases, and cytotoxins cannot all be eliminated simultaneously[31]. Moreover, genetical modification may also affecting bacterial viability and vesicle production efficiency[16]. Using vesicles derived from protoplasts offers a viable alternative. Current safety experiments showed that sg*Pik3cg*-DHP/DGA-NVs quickly normalized inflammatory markers and neutrophil levels post-administration. In contrast, OMV injections led to elevated serum factors and neutrophil levels across multiple organs, even proving lethal in mice, consistent with previous studies[30,32]. This may be due to the fact that the majority of toxins are lost after the removal of bacterial outer membrane and periplasm. Previous studies have also confirmed the safety and non-pathogenic properties of protoplasts or protoplast-derived vesicles[15,33]. Additionally, the inclusion of PEG-conjugated DSPE during physical extrusion could shield dangerous signals on the vesicle surface and prevent their recognition by the mononuclear phagocyte system, greatly reducing the risk of provoking a systemic inflammatory response. Our cytotoxicity, hemolysis assay and toxicity tests have also demonstrated that protoplast-derived NVs are safer than OMVs. Therefore, using physical extrusion to prepare protoplast-derived NVs not only improves yield but also ensures safety.

Tissue distribution assay demonstrated sg*Pik3cg*-DHP/DGA-NVs were highly enriched in the tumor site and predominantly taken up by TAMs. We believe that the precise TAM targeting ability of these NVs were attributed to the combination of pH-responsive long-circulating polymer (DHP) and galactosamine conjugated ligand (DGA). It has been widely accepted that the PEG chain modification could shield the nanocomplex and enabled them to evade clearance by the mononuclear phagocytosis system[34]. Therefore, DHP decoration could prolong the circulation time of sg*Pik3cg*-DHP/DGA-NVs and these NVs accumulated in the tumor site due to the enhanced permeation and retention (EPR) effect. The excellent pH responsive detachment of PEG fragments from NVs was demonstrated by markedly reduced ratio of RhB$^+$ sg*Pik3cg*-DHP/DGA-NVs after acidic PBS treatment and the non-overlap of RhB-labeled PEG fragment and CFSE-labeled NVs in the acidic tumor tissue, which led to the exposure of galactosamine-decorated NVs surface. Unmodified vesicles from protoplasts have demonstrated some tumor penetration, possibly due to specific surface components. In upcoming research, we aim to identify these components to further boost the vesicles' inherent tumor-targeting abilities. Our previous study demonstrated that MGL, which was highly expressed on TAMs, can facilitate uptake of nano-complex containing galactosamine moieties by macrophage[20]. The current cellular uptake experiments also revealed that DGA decoration could greatly enhance TAM targeting behavior of sg*Pik3cg*-DHP/DGA-NVs through MGL-mediated endocytosis. Interestingly, intracellular distribution data further demonstrated that NVs could successfully escape from the lysosome after macrophages endocytosis. It is possible that NVs contain some types of bacterial polypeptides, which might be protonated and formed amphiphilic α-helices with high affinity for lysosomal membranes. These α-helices adhere and insert into lysosomal membranes to form pores that help NVs escape from lysosomes[35,36]. Then, CpG-enriched genomic DNA fragments bind to intracellular TLR9 and our luciferase assay, western blotting and cytokine assay demonstrated their immunostimulant effects. Meanwhile, Cas9 protein with nuclear localization sequence enters the nucleus to perform its gene editing function in TAMs, which was confirmed by T7E1 analysis and NGS. Therefore, our triple targeting strategy comprised of a long-circulating polymer, TAM targeting ligand and NVs-unique bacterial polypeptides resulted in a higher gene editing efficiency or protein expression inhibition efficiency than previous reports[37,38]. However, our in vivo distribution experiments demonstrated that sg*Pik3cg*-DHP/DGA-NVs treatment led to the gene editing in only a small fraction of cells in normal organs such as the liver and spleen. Although we did not specifically assess PI3Kγ protein expression levels and related inflammatory factors in these normal organs post-NV administration, bio-safety assessments based on H&E staining and biochemical indexes suggest that nonspecific editing did not lead to significant inflammation or tissue damage in normal organs. This underscores the safety of our vesicle-based therapeutic approach. In future studies, we plan to investigate whether sg*Pik3cg*-DHP/DGA-NVs mediated-gene editing in normal organs might activate tissue-resident immune cells and trigger inflammatory responses. In brief, protoplast-derived NVs have versatile applications, serving as integrated platforms for the production, loading, and modification of a variety of biotherapeutic molecules, including nucleic acids and proteins. Additionally, these bacterial vesicles offer two layers of customization: genetic manipulation of the parent bacteria and surface alteration of the vesicles themselves. This flexibility could address some drawbacks of non-viral carriers like lipid nanoparticles (LNPs), such as drug inactivation and limited targeting capabilities[39].

Previous studies have found that targeting PI3Kγ signaling via siRNA or kinase inhibitor could switch the immunosuppressive TAMs towards immune-stimulatory phenotype and synergize with immune checkpoint inhibitors to suppress tumor growth in various murine cancer models[40,41]. These suggest that specific inhibition of PI3Kγ in

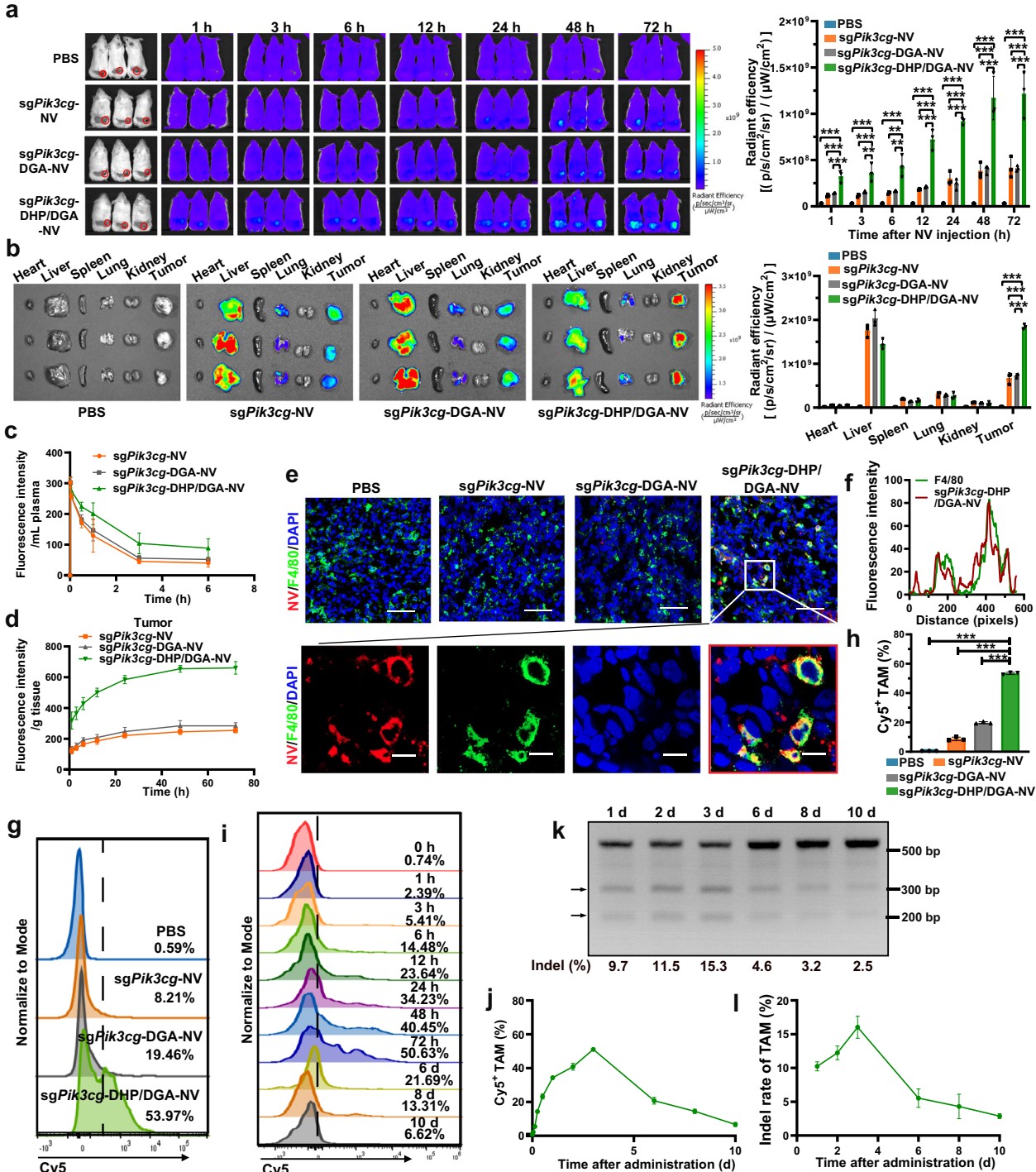

**Fig. 6 | In situ genetic reprogramming and repolarization of TAM in 4T1 tumor-bearing mice via sg*Pik3cg*-DHP/DGA-NVs. a** Fluorescence images of the 4T1 tumor-bearing mice and quantitative analysis of tumor site fluorescence (red circle) at indicated time points after injection of different types of Cy5-NVs (dose: $1 \times 10^{10}$ NVs per mouse). $n = 3$ mice per group. **b** Organ tissues from mice treated with the above-mentioned NVs at 72 h were photographed, and fluorescence intensity was analyzed. $n = 3$ mice per group. **c, d** Cy5 fluorescence intensity in plasma and tumor of mice with above-mentioned NVs treatment was detected at indicated time points (Em:644 nm; Ex: 665 nm). $n = 3$ mice per time point. **e** Immunofluorescence staining of tumor sections harvested from mice at 72 h after Cy5-sg*Pik3cg*-DHP/DGA-NVs injection. Red, Cy5-NVs; green, F4/80; blue, DAPI nuclear staining. Scaled bar = 50 µm for 400× and 10 µm for amplification. The representative images presented are from a sample size of $n = 3$ mice. **f** Fluorescence co-localization analysis of Cy5-NVs

and TAMs using image J software. Fluorescence intensity profile representing the value indicated by the red rectangle. **g, h** Flow cytometry was used to analyze the ratio of Cy5+ TAMs from mice at 72 h after different types of NVs injection (dose: $1 \times 10^{10}$ NVs per mouse). $n = 3$ mice per group. **i, j** The ratio of Cy5+ TAMs at different time points after treatment with $1 \times 10^{10}$ sg*Pik3cg*-DHP/DGA-NVs measured by flow cytometry. $n = 3$ mice per time point. **k, l** T7E1 analysis for indel formation in TAMs from mice at different time points after treatment with sg*Pik3cg*-DHP/DGA-NVs. A representative image is presented. $n = 3$ mice per time point. Data are presented as the means ± SD. Statistical analyses were performed using one-way ANOVA with Dunnett's multiple comparison test. *$P < 0.05$, **$P < 0.01$ and ***$P < 0.001$, ns, no significant change. The exact *P*-value and source data are provided as a Source Data file.

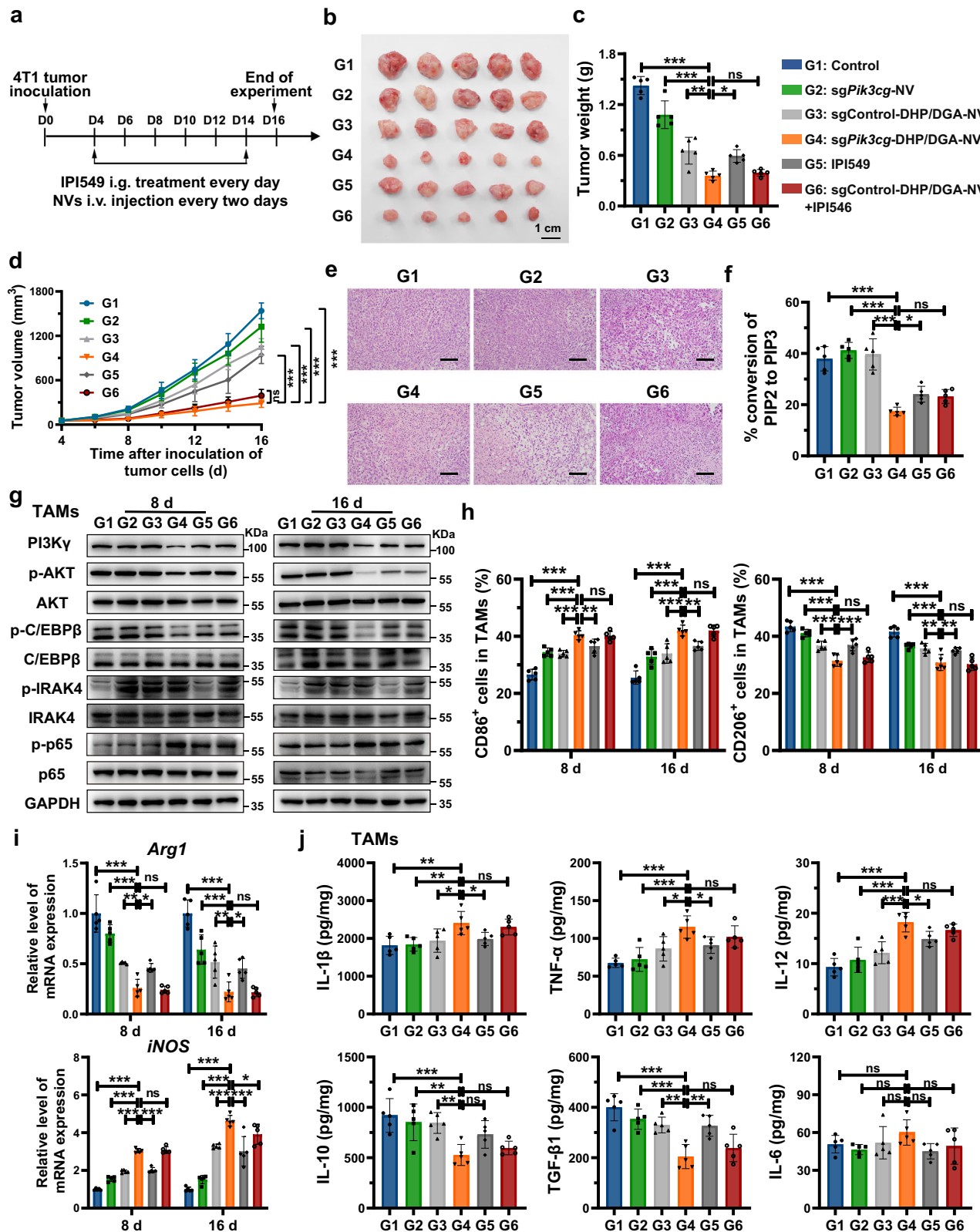

TAMs holds great promise in cancer immunotherapy. However, designing TAM-reprogramming therapies based on PI3Kγ blockade still faces great challenges, particularly in maintaining re-educated TAMs in an anti-tumor immunostimulatory phenotype for prolonged periods. To overcome these challenges, the present study utilized *E. coli* protoplast-derived NVs to deliver Cas9-sg*Pik3cg* RNP into TAMs. This approach

demonstrated superior biosafety and target specificity, exhibiting little influence on the fitness of TAMs. More importantly, the results showed that sg*Pik3cg*-DHP/DGA-NVs treatment exhibited better TAM repolarizing efficiency and therapeutic effects than IPI549. The tumor growth inhibition (TGI) ratio of sg*Pik3cg*-DHP/DGA-NV ranged from 70.5% to 81.8% in the 4T1 mouse tumor model based on the outcomes of four

**Fig. 7 | Anti-tumor immunotherapy was achieved through systemic injection of TAM-targeted sg*Pik3cg*-DHP/DGA-NVs in 4T1 tumor-bearing mice. a** Schematic diagram of 4T1 tumor-bearing mice with in-vein injections of NVs ($1 \times 10^{10}$ NVs every two days) or intragastric administration of IPI549 (15 mg/kg every day). **b, c** Tumor images and tumor weights in mice treated with different NVs or IPI549 on day 16 post-tumor cell inoculation. **d** Tumor volume changes in various treatment groups of 4T1 tumor-bearing mice. **e** Representative H&E staining images of 4T1 tumor sections. Scale bar, 100 μm. **f** Phosphoinositide 3-kinase activity in TAMs, characterized by the conversion ratio of PIP2 to PIP3, was determined by ELISA on day 16 after tumor model establishment. $n = 5$ mice per group for (**b–f**). **g** Levels of PI3Kγ and TLR9 pathways related molecules (p-C/EBPβ, p-AKT, p-IRAK4 and p-p65) in TAMs isolated from mice receiving the mentioned treatments, detected by

western blotting on day 8 and 16 post-tumor cell inoculation. The experiments were repeated three times independently with similar results. **h, i** Effects of sg*Pik3cg*-DHP/DGA-NVs on TAM phenotype were determined by flow cytometry analysis (CD86 and CD206) and qRT-PCR assay (*iNOS* and *Arg1*). $n = 5$ mice for each time point. **j** Cytokine levels in TAMs from 4T1 tumor-bearing mice with different treatments were measured on day 16 after tumor model establishment. $n = 5$ mice per group. Data are presented as the means ± SD. Two-way ANOVA with Dunnett's multiple comparison test was used in (**d**). Other statistical analyses were performed using one-way ANOVA with Dunnett's multiple comparison test except the ELISA assay of IL-6 in (**j**) (Kruskal-Wallis test with Dunn's multiple comparisons test). *$P < 0.05$, **$P < 0.01$ and ***$P < 0.001$, ns, no significant change. The exact $P$-value and source data are provided as a Source Data file.

experiments. In the MC38 mouse tumor model, it was 76.4%, as determined from a single experiment. These results indicate a consistently low variability in the anti-tumor effects mediated by sg*Pik3cg*-DHP/DGA-NVs, highlighting its promising potential in cancer immunotherapy. In our study, IPI549 showed a tumor growth inhibition (TGI) ratio of 39.5% to 51.5% in the 4T1 model from two experiments, and 45.1% in the MC38 model from one experiment, consistent with results from prior[41]. Additionally, we analyzed the TGI ratios of other PI3Kγ antagonists, which exhibited TGIs ranging from 21.2% to 44.1% in the 4T1 tumor model and around 40.0% in the MC38 tumor model[42–45]. Notably, the tumor-suppressive efficacy of sg*Pik3cg*-DHP/DGA-NVs surpasses that of standard PI3Kγ antagonists. This enhanced therapeutic efficacy may be attributed to the permanent inhibition of PI3Kγ activity in TAMs through *Pik3cg* gene knockout using the CRISPR-Cas9 system, in contrast to the short-term inhibition offered by small molecule compounds. Western blotting and enzyme activity assays revealed that sg*Pik3cg*-DHP/DGA-NVs more effectively disrupted the PI3K pathway than IPI549. Moreover, successfully edited TAMs remained viable 10 days post-injection with sg*Pik3cg*-DHP/DGA-NVs. Therefore, sg*Pik3cg*-DHP/DGA-NVs treatment can permanently reshape TAMs into an anti-tumor M1-like phenotype regardless of the immunosuppressive tumor microenvironment. The study also found that bacteria-derived genomic DNA fragments containing CpG-rich sequences were effectively transferred into macrophages by sg*Pik3cg*-DHP/DGA-NVs. These DNA fragments are ideal activators of TLR9 and synergize to induce polarization of macrophages towards an M1-like phenotype by targeting the downstream transcription factor such as NF-κB[46]. Our hypothesis was validated by data showing that the use of TLR9-knockout mice and the TLR9-signaling inhibitor ODN2088 partially neutralized the ability of sg*Pik3cg*-DHP/DGA-NVs to repolarize macrophages and their therapeutic impact. In contrast, co-treatment with sgControl-DHP/DGA-NV and IPI549 exhibited the therapeutic effects comparable to those of sg*Pik3cg*-DHP/DGA-NVs alone. Therefore, the CRISPR-Cas9 system targeting *Pik3cg* and the TLR9 ligands in sg*Pik3cg*-DHP/DGA-NVs are crucial for their potent immune-stimulatory effects on TAMs. TAMs re-educated by sg*Pik3cg*-DHP/DGA-NVs exhibited an activated immune phenotype, characterized by reduced secretion of immunosuppressive cytokines like IL-10 and TGF-β, and increased levels of immunostimulatory cytokines such as IL-12 and TNF-α. These cytokines are pivotal in activating dendritic cells (DCs) and subsequent T-cell activation and proliferation, thereby transforming "cold" tumors into "hot" ones[47]. Both flow cytometry and transcriptomic sequencing confirmed that sg*Pik3cg*-DHP/DGA-NVs successfully shifted the tumor microenvironment towards immune activation, which enhances T cell-mediated anti-tumor efficacy. This treatment also reduced TGF-β level in the tumor tissue, implicating a potential disruption in the activation of tumor-associated fibroblasts and hindrance in the formation of a fibrotic barrier within the tumor[48]. Thus, the durable reprogramming of TAMs can dismantle both the immunosuppressive and fibrotic barriers in the tumor microenvironment, greatly improving the efficacy of various therapeutic agents, including chemotherapy drugs, immunotherapeutics, and cell-based therapies. The

synergistic anti-tumor effects observed with sg*Pik3cg*-DHP/DGA-NVs and PD-1 antibody treatments offer a promising avenue for future combination therapies in oncology.

In conclusion, we developed an in vivo gene-editing technology using bacteria-derived NVs for the efficient delivery of the CRISPR-Cas9 system. This approach enables the effective reprogramming of endogenous TAMs, resulting in significant therapeutic benefits for cancer treatment without triggering systemic immune responses. Our findings suggest that this strategy has potential as a valuable tool in cancer therapy.

## Methods
### Ethical statement
This research complies with all relevant ethical regulations. All animal experimental procedures were conducted in accordance with the National Institutes of Health Guide for the Care and Use of Laboratory Animals and were approved by the Animal Ethical Board of Nanjing University (IACUC-2005005-2).

### Chemical and biological reagents
1,2-distearoyl-sn-glycero-3-phosphoethanolamine-N-Succinimidyl Ester (DSPE), mPEG$_{2000}$-OH and Boc-NH-PEG$_{2000}$-NH$_2$ were purchased from Ruixi Biological Technology CO., LTD (Xi'an, China). Isopropyl β-D-1-thiogalactopyranoside (IPTG), the BCA protein assay kit, PBS, and the *E. coli* BL21 (DE3) strain were obtained from Sangon Biotech (Shanghai, China). IPI549, purchased from MedChemExpress LLC (Shanghai, China), was dissolved in a 5% 1-methyl-2-pyrrolidinone solution in polyethylene glycol 400 for in vivo use. Other reagents were sourced from Sigma-Aldrich (St. Louis, MO, USA). 4T1 cells (catalog number (cat. no.): SCSP-5056), Raw 264.7 cells (cat. no. SCSP-5036), 293 T cells (cat. no. SCSP-502) were obtained by Cell Bank, Chinese Academy of Sciences (Shanghai, China). MC38 cells (cat. no.: 1101MOU-PUMC000523) was obtained from the Cell Resource Center, Peking Union Medical College (Beijing, China).

### Mouse tumor model
Female BALB/c mice and C57BL/6 J mice were procured from Beijing Vital River Laboratory Animal Technology Co. Ltd (Beijing, China). C57BL/6Smoc-*Tlr9$^{em1Smoc}$* (TLR9 KO) mice were obtained from Shanghai Model Organisms Center Inc. (Shanghai, China). All animals were housed in a specific pathogen-free (SPF) environment with 21 ± 2 °C and a relative humidity of 55 ± 10%, with free access to standard food and water. For maximal tumor burden, we complied with the guideline of Animal Ethical and Welfare Committee of Nanjing University and the maximal tumor size did not exceed 2000 mm³. To establish the mouse breast cancer model, 100 μL PBS containing $1 \times 10^6$ 4T1 cells were subcutaneously injected into the right back of 6-week-old female BALB/c mice. For the colorectal cancer model, $1 \times 10^6$ MC38 cells in 100 μL PBS were subcutaneously injected into the right back of 6-week-old female C57BL/6 J mice or 6-week-old female TLR9 KO mice.

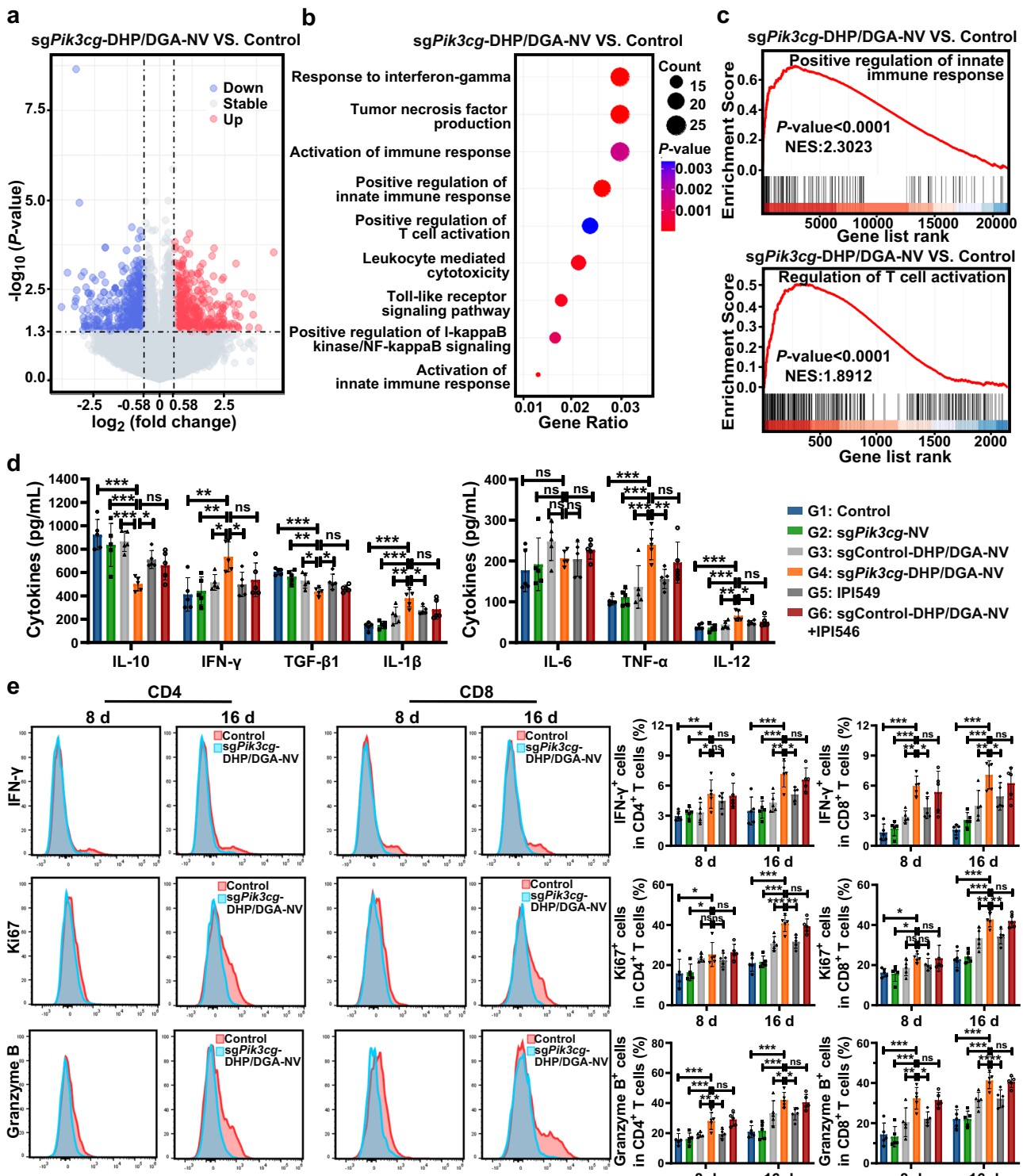

**Fig. 8 | sg*Pik3cg*-DHP/DGA-NVs treatment reshaped the tumor immune microenvironment and induced potent antitumor immunity against breast cancer. a** Volcano plot of the differentially expressed genes (DEGs) based on RNA-seq analysis of 4T1 tumor tissue from the sg*Pik3cg*-DHP/DGA-NVs group ($1 \times 10^{10}$ NVs every two days) compared to the control group on day 16 post-model establishment. $n = 3$ mice per group. **b** Enriched gene ontology (GO) terms associated with DEGs related to immune activation. $n = 3$ mice per group. **c** GSEA enrichment analysis showing the enrichment of genes upregulated in positive regulation of innate immune response and T cell activation. $n = 3$ mice per group. **d** Cytokine levels in 4T1 tumors of mice after treated with either NVs ($1 \times 10^{10}$ NVs every two days) or IPI549 (15 mg/kg every day) were determined by ELISA on day 16 following

the establishment of the tumor model. $n = 5$ mice per group. **e** The influence of sg*Pik3cg*-DHP/DGA-NVs on intratumoral T cell activation and proliferation (percentage of IFN-γ⁺, ki67⁺ and Granzyme B⁺ cells in CD4⁺ T cells and CD8⁺ T cells) of mice with above mentioned treatments was evaluated using flow cytometry analysis at 8 days and 16 days post-tumor cell inoculation. $n = 5$ mice for each time point. Data are presented as the means ± SD. Statistical analyses were performed using one-way ANOVA with Dunnett's multiple comparison test except the ELISA assay of IL-6 in **d** (Kruskal-Wallis test with Dunn's multiple comparisons test). *$P < 0.05$, **$P < 0.01$ and ***$P < 0.001$. ns, no significant change. The exact $P$-value and source data are provided as a Source Data file.

## Preparation and characterization of sg*Pik3cg*-DHP/DGA-NV

*E. coli* harboring CRISPR plasmids (BPK764) were cultured in LB medium at 37 °C. When the OD 600 reached 0.5, isopropyl β-D-1-thiogalactopyranoside (IPTG) was added to induce the expression of the geneome editing system. The bacteria were then cultured overnight at 25 °C with shaking at 200 rpm and finally resuspended in 50 mM Tris-HCl buffer (pH 8.0) containing 20% (w/v) sucrose. Lysozyme (final concentration:1 mg/mL) and EDTA (final concentration: 50 mM) were added to the cell suspension and it was further incubated at 37 °C for 35 min to obtain protoplasts. The protoplasts were washed three times with the fresh Tris-HCl buffer to remove the remaining outer membrane components[49].

The synthesis protocols for a pH-responsive phospholipid derivative (DHP) and a TAM-targeted phospholipid derivative (DGA) were provided in the supplementary information. To create protoplast-derived NVs, protoplasts and DHP/DGA were transferred into a mini-extruder (Avanti Polar Lipids, Birmingham, AL, USA) and passed through 5, 1, and 0.4 µm polycarbonate membrane filters (Whatman, Maidstone, UK) sequentially. The crude extrudate was further purified using iodixanol density-gradient ultracentrifugation (100,000 × g for 2.5 h at 4 °C), and NV fractions were collected from the interface between 50 % and 10 % iodixanol (Axis Shield Diagnostics Ltd, Dundee, Scotland). Finally, the NVs were washed twice, resuspended in PBS, and stored at −80 °C until use. To determine the decoration efficiency of DHP/DGA in NVs, fluorescence-labeled DHP/DGA were added during the extrusion process and the collected NVs were used for flow cytometer and fluorescence resonance energy transfer (FRET) analysis. OMVs were prepared following a previous report[50].

The diameters of NVs after undergoing multiple freeze-thaw cycles, NVs pre-incubated with fetal bovine serum for 24 h, or NVs pre-incubated in PBS for 8 days were examined by Nanosight NS300 (Malvern, United Kingdom). The morphology, Zeta potential and polydispersity index (PDI) of NVs were examined by TEM (JEM-2100, JEOL, Tokyo, Japan), Zetasizer Nano-Z (Malvern) and NanoBrook 90Plus Zeta (Brookhaven, Holtsville, NY, USA), respectively. The yield of NVs was calculated by quantifying the protein content of NVs produced from a known amount of *E. coli*. To assess the pH-responsive ability of DHP, NVs incorporated with RhB-DHP were incubated with PBS (pH 7.4 or pH 6.5) for flow cytometry and FRET analysis. Additionally, 50 µL PBS containing $5 \times 10^9$ CFSE-labeled sg*Pik3cg*-RhB-DHP/DGA-NVs were directly injected into 4T1 tumor tissues and tumor-adjacent tissues, respectively. The tissues were harvested for fluorescence photography, H&E staining and ELISA assay (IL-1β and IL-6) 24 h post-injection.

## Component identification of sg*Pik3cg*-DHP/DGA-NV

To analyze the nucleotide content in NVs, DNA and RNA isolated from *E. coli* and NVs were assessed using LabChip GX (PerkinElmer, Waltham, MA, USA) and Agilent 2100 Bioanalyzer (Santa Clara, CA, USA). The total protein from *E. coli*, *E. coli* protoplasts, OMVs and NVs was extracted and subjected to LC-MS analysis on a Shimadzu UFLC 20ADXR HPLC system connected to an AB Sciex 5600 Triple TOF mass spectrometer (AB SCIEX, Waltham, MA, USA). Subsequently, Max-Quant software was employed to analyze protein mass spectrometry data. The intensity-based absolute quantification (iBAQ) method was applied to rank the abundance of distinct proteins within each group. The protein levels across different samples were compared by label-free quantification (LFQ) intensity, represented by a normalized intensity profile generated using a specific algorithm[51].

The encapsulation of sg*Pik3cg*, Cas9 protein and CpG-rich DNA fragments by NVs was assessed through western blotting and PCR. Absolute quantification of Cas9 protein and sgRNA copies in NVs were conducted using ELISA (Cell Biolabs, Inc, San Diego, CA, USA) and qRT-qPCR, respectively. A luciferase reporter assay was applied to evaluate the stimulatory effect of NVs-encapsulated CpG-rich genomic DNA fragments on the TLR9 pathway. Furthermore, endotoxin content, cell toxicity, hemolysis assay and in vivo toxicity of NVs were performed to examine the biosafety of NVs. Detailed information on the above-mentioned methods is provided in the supplementary information.

## Cellular uptake of NVs in M2-BMDMs and subcellular distribution

0.5 mL of medium containing $5 \times 10^5$ M2-BMDMs was seeded into 24-well plates overnight before the addition of NVs. Then, the medium was removed, and $6 \times 10^8$ CFSE stained sg*Pik3cg*-NV, sg*Pik3cg*-DGA-NV, or sg*Pik3cg*-DHP/DGA-NV treated with either PBS at pH 7.4 or pH 6.5 in 0.5 mL DMEM medium, was introduced into the wells. After incubation at 37 °C for 3 h, the medium containing NVs was removed, and M2-BMDMs were washed twice with PBS before fresh medium was added. Uptake efficiency was examined using a ZEISS confocal microscope (ZEISS LSM 980, Oberkochen, Germany), a flow cytometer (Thermo Fisher Attune, Waltham, MA, USA), and the fluorescence intensity of the treated cell lysate was quantified using a microplate reader (Thermo Fisher, Em: 488 nm; Ex: 530 nm). To investigate the role of MGL-mediated endocytosis in the uptake of DGA-labeled NVs, potential competitors such as GalNAc or GlcNAc (100 mmol/L) were pre-incubated with BMDMs for 1 h prior to adding NVs. Moreover, BMDMs were transfected with siRNA targeting *Mgl*1/2 for 48 h before the addition of NVs.

To investigate the subcellular localization of NVs in M2-BMDMs, CFSE-labeled DHP/DGA-NVs were incubated with M2-BMDMs for 3 h. Subsequently, the cells were stained with LysoTracker Red (Beyotime Biotechnology, Shanghai, China) to visualize lysosomal escape, and observations were made using a ZEISS confocal microscope. The co-localization of NVs and lysosomes was analyzed using ImageJ Software. To assess the nuclear import of Cas9 protein, immunofluorescence staining and western blotting assays were conducted. Additionally, the presence of nuclear sgRNA, total sgRNA, and CpG-rich DNA sequences in M2-BMDMs, brought in by NVs, was characterized 12 h (except CpG-rich DNA sequences for 3 h), after the addition of NVs using PCR amplification and agarose gel electrophoresis. The information of primers and antibodies used was shown in Supplementary Data 10–11.

## Cell treatments

To examine the effects of NVs, $1.5 \times 10^6$ M2-BMDMs were seeded in 6-well plates and allowed to incubate for 24 h prior to the addition of sg*Pik3cg*-DHP/DGA-NVs. These NVs were pre-treated with pH 6.5 PBS, and then were introduced to BMDMs for 6 h, followed by the incubation with fresh medium for 42 h. In certain experimental conditions, M2-BMDMs were treated with ODN1826 (a classical mouse TLR9 agonist, final concentration: 10 µmol/L) and IPI549 (a PI3Kγ inhibitor, final concentration: 1 µmol/L) for 48 h. To further investigate the role of NVs encapsulated CpG-rich DNA sequences in macrophage repolarization, 10 µmol/L ODN2088 (a TLR9 inhibitor) was added to BMDMs 30 min prior to NVs treatment and co-incubated for 48 h. A T7E1 assay was conducted to examine gene-editing efficiency, and next-generation sequencing (NGS) was performed to evaluate induced indel patterns. Western blotting was utilized to determine the protein levels of molecules related to the PI3Kγ and TLR9 pathways. The phenotype of BMDMs was characterized by quantifying mRNA levels of M1/M2 markers (iNOS and Arg1), performing ELISA assays for cytokines, and analyzing M1/M2 markers (CD86 and CD206) using flow cytometry. Detailed methods are provided in the supplementary information.

To assess the persistence of the genome editing system in macrophages, a conditioned medium (CM) was prepared using 4T1 cells. These cells were cultured in serum-free 1640 medium for 24 h, and the resulting supernatant was filtered through a 0.22 µm membrane filter to create the 4T1 cell-CM. Next, $1.5 \times 10^6$ BMDMs in 6-well plates were

treated with $1.8 \times 10^9$ sg*Pik3cg*-DHP/DGA-NVs for 6 h and further cultured in fresh DMEM medium for an additional 18 h. Subsequently, the BMDMs were exposed to 4T1 cell-CM for 24 h. Additionally, another group of BMDMs was treated with 1 μmol/L IPI549 for 24 h, followed by the addition of 4T1 cell-CM containing an equivalent concentration of IPI549 for another 24 h. Finally, the collected macrophages were analyzed for phenotype using RT-qPCR and flow cytometry.

## Biodistribution assay of NVs in mouse tumor models

To establish the most effective dosing strategy for NVs in mouse tumor models, we initially conducted a series of preliminary tests with varied doses and schedules. The optimal regimen was determined to $1 \times 10^{10}$ NVs per mouse, administered bi-daily. This specific dosage and frequency were found to maximize targeting efficiency and gene-editing capability in TAMs. In exploring the biodistribution of NVs within these models, we injected 4T1 tumor-bearing mice intravenously via the tail vein with 100 μL of PBS containing $1 \times 10^{10}$ Cy5-labeled sg*Pik3cg*-NVs, sg*Pik3cg*-DGA-NVs, or sg*Pik3cg*-DHP/DGA-NVs. For control purposes, a group of mice received only PBS. To monitor in vivo distribution following repeated doses, the aforementioned Cy5-labeled sg*Pik3cg*-DHP/DGA-NVs were administered every two days. The IVIS Spectrum system (PerkinElmer) was then employed to capture images of mice and their organs at specified time points. Tumor tissues, other organs and blood were collected for immunofluorescence staining, quantification of fluorescence intensity, and flow cytometry. Additionally, we isolated TAMs to assess gene-editing efficiency at different time points. For detailed methods, please refer to the supplementary information.

## Anti-tumor activity of sg*Pik3cg*-DHP/DGA-NVs

Tumor-bearing mice were randomly assigned to receive 100 μL PBS containing $1 \times 10^{10}$ different type of NVs (sg*Pik3cg*-NVs, sg*Pik3cg*-DHP/DGA-NVs, and sgControl-DHP/DGA-NVs) via tail vein injection every two days. Alternatively, they received IPI549 solution by oral gavage once a day at 15 mg/kg, starting on day 4 after tumor cell inoculation. In some cases, mice were co-treated with other reagents: 1) ODN2088 or its corresponding control (50 μg per mice every two days), 2) anti-TNF-α antibody (500 μg per mouse every two days), 3) anti-CD8α antibody (100 μg per mouse every 3 days), 4) anti-PD-1 antibody (250 μg per mouse every 3 days) via intraperitoneal injection. The relative tumor volume (RTV) = (tumor volume on measured day)/(tumor volume on day 0). TGI ratio was calculated as following described: TGI (%) = [1 − (RTV of the treated group)/(RTV of the control group)] × 100 (%)[52]. Tumor volume was calculated using the formula: (length × width$^2$)/2. Tumor tissues were excised, weighed, and used for H&E staining, immunopathological staining, RT-qPCR assay, transcriptome sequencing, and ELISA assay on day 16 for the 4T1 animal model and day 19 for the MC38 animal model after tumor cell inoculation. Tumor and peripheral blood leukocytes were harvested for flow cytometry. TAMs were purified for ELISA assay and western blotting analysis. Body weight change during treatment, biochemical indicator assay (Jiancheng Bioengineering, Nanjing, China), and histological examination were performed to evaluate the in vivo safety of NVs. Detailed methods are provided in the supplementary information.

## Statistical analysis

Results are expressed as the Mean ± SD. Data were processed in GraphPad Prism 8 software (GraphPad Software Inc. La Jolla, CA, USA) by two-tailed Student's t-test, Mann-Whitney test, one-way ANOVA, two-way ANOVA or Kruskal-Wallis test. Survival rates were analyzed using a survival curve with the Log-rank (Mantel-Cox) test. The exact sample size and statistical test for each experiment are outlined in the corresponding figure legends. Statistical significance was considered when $P < 0.05$, and "ns" indicates no significance.

## Reporting summary

Further information on research design is available in the Nature Portfolio Reporting Summary linked to this article.

## Data availability

The next-generation sequencing data generated in this study have been deposited in the Sequence Read Archive (SRA) repository under accession code PRJNA1018426. RNA sequencing data are available in the Gene Expression Omnibus (GEO)/NCBI public database under accession code GSE243428. The mass spectrometry proteomics data have been deposited to the ProteomeXchange Consortium (http://proteomecentral.proteomexchange.org) via the iProX partner repository with the dataset identifier PXD045507[53,54]. The obtained data of mass spectrometry proteomics was processed and then searched using the integrated Andromeda search engine against the UniProt database for *E. coli* BL21 (DE3) [https://www.uniprot.org/uniprotkb?facets=reviewed%3Afalse&query=BL21%28DE3%29&view=cards]. The remaining data are available within the Article, Supplementary Information or Source Data file. Source data are provided in this paper. Source data are provided with this paper.

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

## Acknowledgements

This work was supported by the National Natural Science Foundation of China (32230058 (J.Z.), 81972267 (Z.H.), 31870821 (J.C.), 31771550 (Z.H.) and 81973273 (J.Z.)).

## Author contributions

Z.H., J.Z., and J.C. designed the experiments, supervised all aspects of study, and contributed to manuscript preparation. M.Z., X.C., P.S., Y.D., L.X., Z.C., X.S., C.G. carried out the experimental work. M.Z., Z.H., X.C., P.S., Y.D., and Y.W. analyzed the data. Z.H., J.C., and J.Z. contributed reagents/materials/analysis tools. M.Z. and X.C. prepared the manuscript. Z.H., J.C., and J.Z. revised the manuscript.

## Competing interests

The authors declare no competing interests.
