## [Peer Review File · Nature Communications]

Bacterial Protoplast-Derived Nanovesicles Carrying CRISPR-Cas9 Tools Re-educate Tumor-Associated Macrophages for Enhanced Cancer ImmunotherapyREVIEWER COMMENTS

Reviewer #1 (Remarks to the Author): with expertise in bacteria inspired nanosystems, cancer therapy

Comments to NC-23-12674

This manuscript reported by Junfeng Zhang et al “Bacterial Protoplast-Derived Nanovesicles Carrying CRISPR Cas9 Tools Efficiently Re-educate Tumor-Associated Macrophages for Cancer Immunotherapy”, the authors developed an innovative in vivo gene-editing technology using bacteria-derived NVs for the efficient delivery of the CRISPR-Cas9 system. This approach enables the effective reprogramming of endogenous TAMs, resulting in significant therapeutic benefits for cancer treatment without triggering systemic immune responses. Our findings suggest that this strategy has great potential as a valuable tool in cancer therapy. In this style of manuscript, it could not consider publish, following points should be answered.

Major points:

1. For the structure issue (in Figure 1), the authors have illustrated that the sgPik3cg-DHP/DGA-NVs are constructed via co-extrusion. I would suggest that the non-covalent interactions of these components should be clarified in the main text, meanwhile, the driving forces for the nanoscale construction and stabilization of sgPik3cg-DHP/DGA-NVs should be specified to elaborate the underlying assembly mechanisms.
- 2 For the components analysis of sgPik3cg-DHP/DGA-NVs issue (Figure 3-e), the ratios of E.coli, E.coli-OMV, E.coli protoplast, sgPik3cg-DHP/DGA-NV were different, how to control the concentration for them to compare them each other in the experiments?
3. For morphological characterization issue, TEM imaging of only two particles is inadequate, the dispersibility and uniformity of sgPik3cg-DHP/DGA-NVs should be displayed and more NVs should be captured during TEM imaging for better representativeness.

Moreover, the particle sizes of NVs and sgPik3cg-DHP/DGA-NVs show no change before and after co-extrusion, please explain.

4. Since sgPik3cg-DHP/DGA-NVs contains two DSPE components whose dynamic behaviors mediate the in vivo performance of the nano-system, I would suggest that the assembly and disassembly of DHP and DGA should be investigated by, for example, FRET measurement.

5. For in vitro study issue (in Figure 4), the fluorescent microscopy images are not very good, I cannot distinguish the internalized NPs from the fluorescence aggregates. Besides, I express my concern, that some subfigures in Figure 4c and Figure 4g are too perfect, for example, there is almost no internalization in the last subfigure of Figure 4c and the fourth subfigure of Figure 4g. It is unusual for in vitro study, in my research experience, the NPs with or without targeting group can be taken up by cells, especially macrophages. So, the authors should explain and discuss this phenomenon in the main text.

6. For in vitro M2 repolarization issue, tumor cells supernatant is used to simulate the immunosuppressive tumor microenvironment. In my opinion, the specific M2-related cytokines produced by tumor cells should be quantified in the tumor cells supernatant, and their functions should be described in the main text.

7. For the biodistribution study issue (in Figure 6), the following issue should note:

(1) In vivo imaging at 72 h post-injection shows fluorescence signals at tumor sites of sgPik3cg-NV and sgPik3cg-DGA-NV groups, but there is no fluorescence at tumor sites for the ex vivo imaging of the mentioned two groups. Why?

(2) I am surprised that the radiant efficiencies for in vivo and ex vivo imaging are in an equal order of magnitude. This is quite unusual.

(3) To claim that "sgPik3cg-DHP/DGA-NVs significantly accumulated in the tumor tissue", the authors should perform the quantification of Cy5 content for tumor and other main tissues.

8. For immunofluorescence study issue (in Figure 6 and Figure 7), the quality of the IF images should be improved and the technical conditions should be equal. What are those shadows in the staining images of Figure 6c?

Minor points:

1 In page 7, in the title of sgPik3cg-DHP/DGA-NVs, what was the “.....M2 macrophages in an in vitro study.....”.

2 In page 7, “ the packaging efficiency.....is 23.00% and 24.46%, respectively”, is would change to “efficienciesare.....,respectively”.

3 pave the way change in to paved the way

4 In page 5, in briefly issue, it was said that 117.5 ug NVs were obtained, in my opinion, it was difficulty to weight the level of ug.

5. For in vivo study, there is no pharmacokinetics study, please supplement.

6. The quality of Figure 2g need to be improved, it looks rather blurry. Why the NPs only distributed in a small region within tumor tissue, but showed uniform distribution in the tumor-adjacent region? Please explain.

Reviewer #2 (Remarks to the Author): with expertise in bacteria inspired nanosystems, cancer therapy

In this manuscript, the authors reported a gene editing delivery system using functionalized nanovesicles (NVs) derived from E. coli protoplast as to highly effectively encapsulate the Cas9-sgRNA RNP, thereby realizing reprogramming of tumor-associated macrophages (TAMs) and in vivo effective tumor treatment. However, the vector construction in the manuscript does not have novelty, and there is fungibility of the bacterial protoplast-derived nanovesicles. On the one hand, the safety of protoplast-derived nanovesicles is emphasized, and acid-responsive PEG is connected to the membrane and CPG is loaded. Here, whether PEG modified OMVs can be directly replaced and realized. On the other hand, sgPik3cg-DHP/DGA-NVs processing is rough and arbitrary for the content of a Cas9-sgRNA RNP and CpG-rich genomic DNA fragments, and perhaps LNP is more suitable as a vector than protoplast-derived nanovesicles. In short, the overall design does not show irreplaceability

and ingenious, is boring. In addition, there is no good data to support the uniformity and stability of modified protoplast derived nanovesicles. Therefore, it is not recommended that the article be published in Nat Commun.

Reviewer #3 (Remarks to the Author): with expertise in cancer immunology, macrophages, PI3K targeting

The manuscript from Zhao et al describes a novel approach to produce and formulate CAS9 – sgPIK3cg containing nanovesicles to modify the function of MDSCs in the tumor microenvironment. The authors show this complex formulation derived from CAS9 – and sgPIK3cg expressing bacteria can modify PIK3cg expression in tumour associated macrophages reducing the differentiation of suppressive macrophages. Systemic administration of the vesicles resulted in reduction of syngeneic tumour growth. Data is provided to suggest the anti-tumour activity is through the reprogramming of macrophages following reduction in PI3K expression but required induction of TLR signalling as a result of the introduction of the bacterial protoblast associated products.

The manuscript is well written in the main and data clearly laid out. This is a complex therapeutic approach therefore providing sufficient insights to understand the primary drivers of efficacy by isolating the contributions of components from each element of the NV formulation in vivo is challenging. While the authors have performed experiments to look at mode of action there are a number of points where the manuscript could be improved.

Points that should be addressed:

It is not clear how the targeting of the TAM/MDSC is achieved. Reference 17 is provided as a justification of the targeting through MGL but this is a review not a primary data manuscript. This reviewer is not as familiar with this targeting strategy with the galactosamine for targeting tumour or MDSC specifically. Some data to show there is a bias to this uptake mechanism in tumour versus normal (tissue) macrophages or other cell types would be helpful.

Throughout the manuscript concentrations or doses used are not clear for any reagent used e.g. nanovesicles, IPI549 etc.

Figure 2 The amount of particle injected is not stated in the figure legend. Panel G The intratumoural injection is a rather challenging experiment with significant amount of material injected to tumour which in itself can induce inflammation and tissue damage. To allow a better understanding of the experiment a high level image of the tumour should be shown that together encompass the tumour and associated normal tissue distribution to allow the relative signal levels to be compared, as well as the level of tissue damage to be assessed. Panel H the changes in peripheral cells should be shown, are there changes in neutrophil levels etc in the peripheral blood, and is there evidence of immune cell changes in liver or spleen indicating systemic exposure from the injections. If not it is not clear how this experiment demonstrates that the NV is well tolerated?

There is no data in the paper showing pathway modulation by the KO of PI3K α , or the induction of TLR9 or other inflammatory signalling pathways in the tumour, normal cells or macrophages. All the data are by inference with outcome biomarkers such as CD206/CD86, or cytokine production being taken as indicative of pathway modification. Evidence of signalling changes (PI3K α associated, TNF, IFN, IL6 etc.) should be provided in the in vitro experiments. Modification of PI3K α signalling should be compared to IPI549.

The in vivo reduction of PI3K α protein appears very modest from the data shown. Does this imply there is not good modification of PI3K α across the macrophage population? The poor modification of PI3K α protein levels may be enough but a comparison with IPI549 modification of monocyte / macrophage function through PI3K α signalling should be performed to establish the relevance of the effect. This is important as there is an impact of the TLR signalling and the balance of TLR vs PI3K α in the context of the therapeutic effect of the NV is not clear throughout the studies. The authors have tried to address this with the use of a TLR antagonist compound but it does not provide enough data to determine whether modification of the PI3K α mediated differentiation program is a major or minor driver of the effect.

Where IPI549 is used as a comparator in vivo dosing should be maintained compared to the NV. For example in Figure 5 it seems that IPI549 was only dosed for 6 hours before cells were transferred to new media, the legend that the cells were then cultured without IPI549. A fair comparator would be to continue the incubation with IPI549 following transfer to new media.

In Figure 6 the legend implies IPI549 was given as one dose and not continually through the 48hr time course of the experiment. Experiments should be repeated with continuous IPI549 as a control. The rendering of the fluorescence images in Panel b do not reflect the quantification in the graph. It is also surprising to see such marked differences in biodistribution to liver and tumour using this approach (particularly with repeated dosing). The images of the tumour distribution in the right hand Panel of B appear odd compared to other panels. It would be expected to see more in the tumour capsule, does the large red signal in the tumour indicate presence of necrosis? Shown more detailed images would be aid interpretation of this and other similar data. Some images are shown are Figure 7 panel but these are only selected fields and the IHC has not been extensively quantitated.

Is there editing of cells in the bone marrow or peripheral blood? There is little data showing editing of cells in other organs e.g. macrophage from the liver, spleen bone marrow. Are other cell types edited e.g. epithelial cells, endothelial cells.

There is little broader immune cell biomarker data shown apart from CD8 accumulation Figure 8 (where the duration of treatment is not stated). It is not clear what the effect of the treatments are on other cells, particularly immune cells. For example is there a change in the neutrophils in the peripheral blood or tissues (neutrophils also depend on PI3Kg for activation). More comprehensive biomarker assessment would be important given the dual MOA of the formulation.

A combination experiment with a PD1 inhibitor should be performed to confirm claims of the potential to enhance immunotherapy. This should be benchmarked versus IPI549.

Does depleting CD8 T-cells block efficacy in the 4T1 or CT26. This is an important control to confirm the mode of action as being T cell mediated.

Efficacy studies are performed with repeated administration. Does blocking TNF or IL6 have an impact on efficacy achieved with repeated administration? Are dendritic cells activated with repeated administration of NV?

A control experiment of continuous IPI549 treatment with the sgControl – DHP/DGA-NV should be performed to baseline the efficacy derived from the contribution of PI3Kg + the independent NV constituents to treatment.

What fraction of cells are edited in each in vivo administration? How long do edited cells persist for? Is the editing short lived? A time course is required to determine the utility of the approach. This time course should incorporate normal and tumour tissues.

Is there accumulation of the NV over time with the repeated administration? No PK is shown.

A time course of modification of PI3Kg signalling versus activation of other pathways should be shown. For example in the experiment shown in Figure 7 what is the extent of pathway modulation for TNF, STAT, IFN signalling etc. in addition to downstream impact on PI3Kg mediated pathways.

Reviewer #4 (Remarks to the Author): with expertise in cancer immunology, macrophages, PI3K targeting

This is an excellent manuscript detailing the development of a new in vivo Crispr tool. Using bacterial protoplasts bearing a Cas9-sgRNA RNP, the authors deliver Crispr components first to cells and then to tumors. Taken up by macrophages, the protoplasts transfect macrophages with PI3Kgamma siRNA. This alters macrophages by stimulating expression of pro-inflammatory factors that lead to inhibition of tumor growth.

This work is a significant development in the delivery of Crispr to cells in vivo and a rewarding demonstration that PI3Kg inhibition remains a valuable way to suppress tumor growth.

The questions raised by the reviewers are responded point by point as follows:

Reviewer 1:

General comments:

- 1. For the structure issue (in Figure 1), the authors have illustrated that the sgPik3cg-DHP/DGA-NVs are constructed via co-extrusion. I would suggest that the non-covalent interactions of these components should be clarified in the main text, meanwhile, the driving forces for the nanoscale construction and stabilization of sgPik3cg-DHP/DGA-NVs should be specified to elaborate the underlying assembly mechanisms.**

Response: Thanks for the reviewer's suggestions. We clarified the nature of non-covalent interactions in sgPik3cg-DHP/DGA-NVs in the revised manuscript. DHP and DGA are phospholipid derivatives with a polar phosphate head group and two hydrophobic tails. During physical extrusion of protoplasts in the presence of these amphiphilic molecules, hydrophobic tails of DHP and DGA align with those of the disrupted protoplast membrane due to non-covalent, primarily hydrophobic, interactions. This alignment facilitates the formation of self-sealing bilayers in aqueous solutions, termed sgPik3cg-DHP/DGA-NVs. These bilayers present a hydrophilic exterior and a hydrophobic interior, contributing to vesicular stability and functionality. Differential scanning calorimetry (DSC) revealed a phase transition peak specific to sgPik3cg-DHP/DGA-NVs, confirming the successful incorporation of DHP and DGA (Supplementary Fig. 6b). Stability assays further demonstrated that sgPik3cg-DHP/DGA-NVs maintain structural integrity and consistent diameter under various stress conditions, including eight days in PBS, three freeze-thaw cycles, and a 24-hour incubation in serum-containing PBS (Supplementary Fig. 7). This robust stability is attributed to the self-sealing lipid bilayers, which minimize the energetically unfavorable exposure of hydrophobic tails to the aqueous environment.

- 2. For the components analysis of sgPik3cg-DHP/DGA-NVs issue (Figure 3-e), the ratios of E.coli, E.coli-OMV, E.coli protoplast, sgPik3cg-DHP/DGA-NV were different, how to control the concentration for them to compare them each other in the experiments?**

Response: To enable a meaningful comparison in the experiments, we used MaxQuant software to analyze the protein mass spectrometry data. Protein levels

across different samples were quantified by calculating the area under the curve for MS1-level peptide peaks. Consequently, the top 200 abundant proteins in each sample and their cellular localization ratios are presented in Figure 3e and Supplementary Tables 4-7. The results of Supplementary Figure 13 reveals that the abundance of classic outer membrane proteins and periplasmic proteins (e.g., OmpA and DcrB) in *sgPik3cg*-DHP/DGA-NVs was lower than those in OMVs, consistent with the intra-group abundance ranking results.

- 3. For morphological characterization issue, TEM imaging of only two particles is inadequate, the dispersibility and uniformity of *sgPik3cg*-DHP/DGA-NVs should be displayed and more NVs should be captured during TEM imaging for better representativeness. Moreover, the particle sizes of NVs and *sgPik3cg*-DHP/DGA-NVs show no change before and after co-extrusion, please explain.**

Response: Thank you very much for your advice. Following your suggestion, we captured TEM images at lower magnification to observe more NVs, providing a better understanding of their dispersibility and uniformity as shown in Figure 2b of the revised version. Additionally, we characterized NVs using nanoparticle tracking analysis (NTA) and dynamic light scattering (DLS) (Figure 2c and Supplementary Table 1-2). These results confirm that both free NVs and *sgPik3cg*-DHP/DGA-NVs exhibit uniform nano-sized lipid-bilayered vesicular structures with a low polydispersity index (PDI) and average diameters of 152.20 ± 2.93 nm and 157.92 ± 4.29 nm (determined by DLS), respectively.

Furthermore, the incorporation of DHP and DGA into free NVs led to a slight increase in diameter, which can be attributed to the PEG modification, resulting in a minor size increment of the vesicles. However, there was no significant difference in size between NVs and *sgPik3cg*-DHP/DGA-NVs. This could be due to the consistent extrusion pressure applied during the extrusion process, resulting in similar particle sizes.

- 4. Since *sgPik3cg*-DHP/DGA-NVs contains two DSPE components whose dynamic behaviors mediate the in vivo performance of the nano-system, I would suggest that the assembly and disassembly of DHP and DGA should be investigated by, for example, FRET measurement.**

Response: As suggested, we conducted fluorescence resonance energy transfer

(FRET) measurements to investigate the assembly and disassembly of DHP and DGA within NVs. In Supplementary Figure 4a and 4c, DHP and DGA were labeled with Rhodamine B (RhB) and FITC, respectively. In Supplementary Figure 6a, DHP/FITC-DGA-decorated NVs exhibited a strong emission peak at 520 nm with a corresponding excitation wavelength of 488 nm. Meanwhile, RhB-DHP&FITC-DGA-decorated NVs did not exhibit a significant emission peak at 520 nm under the same excitation wavelength (488 nm). After incubation with acidic PBS (pH 6.5) for 24 h, the emission peak at 520 nm of RhB-DHP&FITC-DGA-decorated NVs was restored, indicating the detachment of the RhB-PEG₂₀₀₀ fragment from NVs (Supplementary Figure 8b). In contrast, the spectrum of RhB-DHP&FITC-DGA-decorated NVs incubated with PBS (pH 7.4) for 24 h showed no significant change compared to NVs without any treatment. These FRET results demonstrate the excellent pH-responsive detachment of PEG₂₀₀₀ fragments from NVs.

- 5. For in vitro study issue (in Figure 4), the fluorescent microscopy images are not very good, I cannot distinguish the internalized NPs from the fluorescence aggregates. Besides, I express my concern, that some subfigures in Figure 4c and Figure 4g are too perfect, for example, there is almost no internalization in the last subfigure of Figure 4c and the fourth subfigure of Figure 4g. It is unusual for in vitro study, in my research experience, the NPs with or without targeting group can be taken up by cells, especially macrophages. So, the authors should explain and discuss this phenomenon in the main text.**

Response: We appreciate the reviewer's constructive comments. In response to the reviewer's suggestion, we conducted additional experiments to more accurately assess the internalization of NVs by macrophages. In these added experiments, we used fluorescent antibodies (F4/80) to label the surface of macrophages, and dual-fluorescence microphotographs were captured to precisely examine the effective entry of NVs into macrophages. Additionally, the new images were captured with higher sensitivity settings of the detector (Figure 4c and 4g). Therefore, the revised version's fluorescence images clearly demonstrate that a portion of NVs enters the macrophages, as shown in the last subfigure of Figure 4c and the fourth subfigure of Figure 4g in the revised version. It is noteworthy that flow cytometry results reveal that 23.68% of the cells were

CFSE⁺ in the experimental group of *sgPik3cg*-NVs without targeting group. It is possible that the generalized internalization mechanisms, such as membrane fusion, macropinocytosis or lipid-raft-mediated uptake, may promote the uptake of *sgPik3cg*-NVs by macrophages (*J Extracell Vesicles*. 2014; 3: 10.3402/jev.v3.24641). The content has been added in the revised version at page 8, line 225.

- 6. For in vitro M2 repolarization issue, tumor cells supernatant is used to simulate the immunosuppressive tumor microenvironment. In my opinion, the specific M2-related cytokines produced by tumor cells should be quantified in the tumor cells supernatant, and their functions should be described in the main text.**

Response: We appreciate the reviewer's suggestion. As depicted in Supplementary Figure 21a, we were able to detect TGF- β , IL-33, IL-4, and IL-10 in the 4T1 conditioned medium. These immunoregulatory cytokines from 4T1 tumor cells play a pivotal role in inducing macrophage polarization towards the M2 phenotype and significantly contribute to the creation of an immunosuppressive tumor microenvironment (*Signal Transduct Target Ther*. 2021;6(1):127). Consequently, we have included a description of these cytokines, including TGF- β , IL-33, IL-4, and IL-10 at Page 10, line 280 of the revised manuscript.

- 7. For the biodistribution study issue (in Figure 6), the following issue should note:**

(1) In vivo imaging at 72 h post-injection shows fluorescence signals at tumor sites of *sgPik3cg*-NV and *sgPik3cg*-DGA-NV groups, but there is no fluorescence at tumor sites for the ex vivo imaging of the mentioned two groups. Why?

(2) I am surprised that the radiant efficiencies for in vivo and ex vivo imaging are in an equal order of magnitude. This is quite unusual.

(3) To claim that "*sgPik3cg*-DHP/DGA-NVs significantly accumulated in the tumor tissue", the authors should perform the quantification of Cy5 content for tumor and other main tissues.

Response: There might be some points of confusion regarding the biodistribution study, and we appreciate the opportunity to address these issues. After a thorough review of the original images, we sincerely apologize for the inadvertent use of

images from mice that received different dosages, which resulted in inconsistencies in vesicle distribution and the lack of variation in radiant efficiencies in both the *in vivo* live mouse imaging and *ex vivo* organ fluorescence images. In response to this matter, we conducted a meticulous reevaluation of vesicle distribution, utilizing the same cohort of mice for both live imaging and organ imaging studies. The current findings (Figure 6a-b) undeniably demonstrate a substantial accumulation of *sgPik3cg*-DHP/DGA-NVs within the tumor site, while the *sgPik3cg*-NV and *sgPik3cg*-DGA-NV treated groups exhibit modest vesicle accumulation. To further quantify vesicle biodistribution, tissues were collected from various mice as suggested by the reviewers. These tissues were then homogenized and subjected to fluorescence quantification analysis (Figure 6d and Supplementary Figure 22a). The consistent results align with the *in vivo* imaging and organ photographs, confirming the significant presence of NVs within the tumor tissue of the *sgPik3cg*-DHP/DGA-NVs treatment group.

- 8. For immunofluorescence study issue (in Figure 6 and Figure 7), the quality of the IF images should be improved and the technical conditions should be equal. What are those shadows in the staining images of Figure 6c?**

Response: In accordance with the reviewer's guidance, we conducted additional immunofluorescence staining experiments to improve image quality, as depicted in Figure 6e and Supplementary Figure S26b in the revised version. In the previous version, the shaded regions in Figure 6c image represented the tumor's periphery. However, in the updated version, we have adjusted the imaging locations to focus on the central region of the tumor, ensures a more consistent field of view.

Minor points:

- 1. In page 7, in the title of *sgPik3cg*-DHP/DGA-NVs, what was the ".....M2 macrophages in an *in vitro* study.....".**

Response: Thank you for pointing out grammatical errors. We have modified the title as follows in the revised version: "*In vitro* targeted delivery of *sgPik3cg*-DHP/DGA-NVs to M2 Macrophages".

- 2. In page 7, "the packaging efficiency.....is 23.00% and 24.46%, respectively", is would change to "efficienciesare....., respectively".**

Response: According to the reviewer's advice, we have revised this sentence to "The packaging efficiencies of Cas9 and *sgPik3cg* in *sgPik3cg*-DHP/DGA-NVs

are 23.00% and 24.46%, respectively” at Page 7, line 205 of the revised manuscript.

3. pave the way change in to paved the way.

Response: As suggested, we have modified this sentence to “This advance in *in vivo* CRISPR-Cas9 technology opens new avenues for cancer immunotherapy, overcoming challenges related to cell viability and safe, precise *in vivo* delivery” in the revised version.

4. In page 5, in briefly issue, it was said that 117.5 µg NVs were obtained, in my opinion, it was difficulty to weight the level of µg.

Response: We apologize for the previous unclear quantitative description of NVs. In the revised manuscript, as indicated on the page 7, line 210, the descriptions of NVs' quantity have been amended to state, " 1×10^9 CFU bacteria can produce *sgPik3cg*-DHP/DGA-NVs with a total protein amount of 117.5 µg".

5. For in vivo study, there is no pharmacokinetics study, please supplement.

Response: In accordance with the reviewer's suggestion, we conducted a pharmacokinetics study, and the results are depicted in Figure 6c in the revised version. These results illustrate the extended circulation time of *sgPik3cg*-DHP/DGA-NVs when compared to *sgPik3cg*-NVs and *sgPik3cg*-DGA-NVs.

6. The quality of Figure 2g need to be improved, it looks rather blurry. Why the NPs only distributed in a small region within tumor tissue, but showed uniform distribution in the tumor-adjacent region? Please explain.

Response: Following the reviewer's suggestion, we conducted additional experiments and significantly enhanced the quality of the previous Figure 2g. This improved figure is now presented as Figure 2f in the revised manuscript. Figure 2f was captured at lower magnification to include both the tumor and tumor-adjacent areas in the same field, as shown in Figure 2f and Supplementary Figure 9. From the results in Figure 2f, it is evident that within the tumor region, there is not a complete overlap between the red fluorescence-labeled PEG₂₀₀₀ fragments and the green fluorescence representing NVs. This suggests the breakage of the hydrazone bond of DHP in the acidic 4T1 tumor tissue. However, in the tumor-adjacent areas, the distribution of these two fluorescence signals is completely overlapped due to the neutral pH environment.

Reviewer 2:

In this manuscript, the authors reported a gene editing delivery system using functionalized nanovesicles (NVs) derived from *E. coli* protoplast as to highly effectively encapsulate the Cas9-sgRNA RNP, thereby realizing reprogramming of tumor-associated macrophages (TAMs) and *in vivo* effective tumor treatment. However, the vector construction in the manuscript does not have novelty, and there is fungibility of the bacterial protoplast-derived nanovesicles. On the one hand, the safety of protoplast-derived nanovesicles is emphasized, and acid-responsive PEG is connected to the membrane and CPG is loaded. Here, whether PEG modified OMVs can be directly replaced and realized. On the other hand, sgPik3cg-DHP/DGA-NVs processing is rough and arbitrary for the content of a Cas9-sgRNA RNP and CpG-rich genomic DNA fragments, and perhaps LNP is more suitable as a vector than protoplast-derived nanovesicles. In short, the overall design does not show irreplaceability and ingenious, is boring. In addition, there is no good data to support the uniformity and stability of modified protoplast derived nanovesicles.

Response: We greatly appreciate your constructive suggestions. In this study, we engineered bacteria to enable high-expression of ribonucleoprotein (RNP) for *in vivo* gene editing. Using a process that includes removing the highly toxic outer membrane, incorporating targeting ligands, and applying continuous extrusion and filtration techniques, we successfully obtained TAM-targeting nanovesicles loaded with high amounts of RNP. This biovesicle platform offers integrated solutions for producing, loading, and modifying various biotherapeutic molecules, including both nucleic acids and proteins. Importantly, our approach addresses two major challenges associated with bacterial natural vesicles (e.g., OMVs): low production efficiency and high toxicity. Additionally, it overcomes limitations of non-viral carriers like lipid nanoparticles (LNPs), such as potential drug inactivation and limited targeting options. In the revised version, we have conducted experiments to assess the loading efficiency and safety of sgPik3cg-DHP/DGA-NVs, which unlike LNPs, offer a simplified preparation process, effective RNP preservation, and diverse targeting strategies. Details are elaborated in the revised version.

A major hurdle in developing extracellular vesicle-based drug delivery systems is the difficulty in large-scale, homogeneous vesicle production (*Adv Drug Deliv*

Rev. 2021;173:252-278). In comparison, 1×10^9 CFU bacteria can yield sgPik3cg-DHP/DGA-NVs with a total protein content of 117.5 μg , far surpassing the 5.42 μg of OMVs produced after a 24-hour cultivation period. Physical extrusion and filtration techniques for vesicle preparation are widely accepted (*Adv Healthc Mater.* 2022;11(19):e2200142) and achieve much higher encapsulation efficiencies of 23.00% for Cas9 and 24.46% for sgPik3cg compared to approximately 0.87% and 0.54% in OMVs (Supplementary Fig. 14a). The level of Cas9 and sgPik3cg in sgPik3cg-DHP/DGA-NVs is 26 times higher than those in OMVs when cultivated from 1×10^9 CFU bacteria (Supplementary Fig. 14b). Furthermore, OMVs are susceptible to various environmental factors like temperature, antibiotic exposure, and culture density variations, affecting both their yield and protein composition (*Nat Rev Microbiol.* 2019;17(1):13-24). As a result, OMV production requires larger bacterial cultures, extended cultivation times, and stringent conditions, escalating production costs and hindering clinical applications. These findings underscore the advantages of protoplast-derived vesicles for large-scale production.

In addition to the challenges of large-scale production, the safety of using OMVs as delivery vehicles must also be emphasized. OMVs, originating from bacterial outer membranes, contain various virulence factors that could cause severe systemic inflammation upon injection. This poses a serious risk, especially for cancer patients in fragile health conditions (*Acta Pharmacol Sin.* 2018;39(4):514-533; *Shock.* 2023; 59(2): 161–172). Our supplementary safety experiments showed that injecting OMVs led to significant increases in serum factors and neutrophil levels in multiple organs, even causing lethality in mice, consistent with prior studies (*PLoS One.* 2010;5(6):e11334; *Shock.* 2012;37(6):621-8). Conversely, sgPik3cg-DHP/DGA-NVs derived from protoplasts quickly normalized inflammatory markers and neutrophil levels post-administration (Supplementary Figure 10 d-g). This is likely due to the elimination of most toxins when the bacterial outer membrane and periplasmic space were removed. While genetic modifications targeting lipopolysaccharide (LPS) biosynthesis genes could potentially reduce OMV toxicity, it remains a challenge to concurrently remove other virulence factors like bacterial adhesins, proteases, and cytotoxins (*J Control Release.* 2020;323:253-268).

LNPs are widely used non-viral vectors for delivering gene editing tools like

plasmid DNA, Cas9 mRNA, sgRNA, and RNPs (*Mol Pharm.* 2022;19(6):1669-1686). Of these, RNPs offer advantages such as rapid action, reduced off-target effects, and significantly improved editing efficiency with 10-fold higher target specificity (*Nat Biotechnol.* 2015; 33(1): 73–80). However, obtaining pure, bioactive Cas9 protein is a laborious process, involving intricate steps like protein isolation, purification, and establishing optimal storage conditions (*Acta Pharm Sin B.* 2021;11(8):2150-2171). Additionally, the distinct physicochemical properties of Cas9 and sgRNA, the large size of Cas9 protein (~160 kDa), and the vulnerability of RNPs to denaturation all place stringent requirements on delivery carrier construction, raising production costs (*Biomater Sci.* 2022;10(5):1166-1192). In contrast, our method involves high-efficiency expression of gene editing tools in bacteria and uses a simple continuous extrusion and fragmentation technique to prepare vesicles from protoplasts. This bypasses the need for procedures or chemicals that could denature RNP, thus preserving their activity, which was further confirmed by gene editing efficacy evaluation of *Pik3cg* both *in vitro* and *in vivo* (Figure 5a and Figure 6k).

Furthermore, LNPs naturally target the liver, requiring additional ligand modifications to reach other organs or specific cell populations effectively (*Nat Commun.* 2020;11(1):3232; *Nat Protoc.* 2023;18(1):265-291). In contrast, vesicles from protoplasts have shown some ability to penetrate tumor sites even without modification (Figure 6a-b, 6d). This is likely due to specific surface components on the protoplast-derived vesicles, making further enhancements in tumor targeting a more straightforward task. Finally, from a design perspective, bacterial vesicles offer dual-level modifications: genetic engineering of the parent bacteria and surface alteration of the vesicles themselves. Additionally, bacterial polypeptides carried by these nanovesicles can help RNPs escape lysosomes by forming pores in their membranes (Supplementary Figure 18). This multi-layered approach not only broadens the scope of targeted delivery strategies but also positions protoplast-derived nanovesicles as versatile platforms for delivering a wide array of therapeutic biomolecules to lesion-specific tissue cells.

In response to the reviewer's suggestions, we conducted further experiments to characterize the protoplast-derived vesicles. We captured low-magnification TEM images featuring numerous NVs (Figure 2b) and assessed the morphology of *sgPik3cg*-DHP/DGA-NVs using nanoparticle tracking analysis (NTA) and

dynamic light scattering (DLS) (Figure 2c and Supplementary Table 1-2). These data confirm that *sgPik3cg*-DHP/DGA-NVs are uniform, nano-sized, lipid bilayer structures with an average diameter of 157.92 ± 4.29 nm (determined by DLS). They exhibit a low polydispersity index (0.228 ± 0.009) and zeta potential (-45.7667 ± 1.33 mV), indicating good uniformity and stability. For more stability tests, *sgPik3cg*-DHP/DGA-NVs were subjected to various conditions, including eight days of incubation in PBS, three freeze-thaw cycles, and 24-hour incubation in serum-containing PBS. Their excellent stability was verified using NTA and TEM (Supplementary Figure 7).

Overall, we developed an innovative *in vivo* CRISPR-Cas9 system that specifically targets tumor-associated macrophages (TAMs). Utilizing bacterial protoplast-derived nanovesicles, we achieved precise gene-editing in TAMs. This advancement in CRISPR-Cas9 technology opens new avenues for cancer immunotherapy.

Reviewer 3:

- 1. It is not clear how the targeting of the TAM/MDSC is achieved. Reference 17 is provided as a justification of the targeting through MGL but this is a review not a primary data manuscript. This reviewer is not as familiar with this targeting strategy with the galactosamine for targeting tumour or MDSC specifically. Some data to show there is a bias to this uptake mechanism in tumour versus normal (tissue) macrophages or other cell types would be helpful.**

Response: Thanks for your advice. In the revised version, we have replaced reference 17 by references 18-20 with experimental papers related to MGL receptor-mediated endocytosis and previous research focused on TAM targeting through the use of the macrophage galactose-type lectin (MGL) receptor (*J Biol Chem.* 2002;277(23):20686-93; *Nanoscale.* 2019;11(42):20206-20220 and *J Control Release.* 2012;158(2):286-92). MGL has a specific capacity for recognizing glycans with terminal galactose/N-acetylgalactosamine moieties, serving as an endocytic receptor for glycosylated particles (*Trends Immunol.* 2008;29(2):83-90.). In accordance with the reviewer's suggestions, we selected cell lines with varying levels of MGL expression, including those with high MGL expression, such as M2-BMDM and TAMs, those with moderate expression levels, like peritoneal macrophages (PEM) and M0-BMDM, and those with

minimal MGL expression, such as 4T1 and MC38 tumor cells (Supplementary Figure 15-16). Our observations revealed a positive correlation between the efficiency of sgPik3cg-DHP/DGA-NVs entering various cell types and their respective MGL expression levels (Supplementary Figure 16). Additionally, the downregulation of MGL expression by siRNA resulted in reduced efficiency of sgPik3cg-DHP/DGA-NVs entering M2-BMDM (Supplementary Figure 17c-d). All these results indicate that sgPik3cg-DHP/DGA-NVs can selectively enter TAMs via MGL mediated endocytosis.

- 2. Throughout the manuscript concentrations or doses used are not clear for any reagent used e.g. nanovesicles, IPI549 etc.**

Response: We appreciate the reviewer's advice. The detailed concentrations or doses of reagents were added in the figure legends and methods of the revised version.

- 3. Figure 2 The amount of particle injected is not stated in the figure legend. Panel G The intratumoural injection is a rather challenging experiment with significant amount of material injected to tumour which in itself can induce inflammation and tissue damage. To allow a better understanding of the experiment a high level image of the tumour should be shown that together encompass the tumour and associated normal tissue distribution to allow the relative signal levels to be compared, as well as the level of tissue damage to be assessed. Panel H the changes in peripheral cells should be shown, are there changes in neutrophil levels etc in the peripheral blood, and is there evidence of immune cell changes in liver of spleen indicating systemic exposure from the injections. If not it is not clear how this experiment demonstrates that the NV is well tolerated?**

Response: Thanks for the reviewer's advice. The amount of sgPik3cg-DHP/DGA-NVs and OMVs used for *in situ* or intravenous injections were described in the legend of Figure 2.

As suggested, we conducted additional experiments and have included high-level images captured at lower magnification levels, encompassing both the tumor and its adjacent areas in the revised version (Figure 2f). This approach allows for a more comprehensive comparison of signal levels and provides a better assessment of tissue damage. Our findings reveal that within the tumor region, there isn't a complete overlap between the red fluorescence-labeled

PEG₂₀₀₀ fragments and the green fluorescence representing NVs. However, in the tumor-adjacent areas, the two fluorescence signals exhibited complete overlap. Moreover, we performed macroscopic observations, H&E staining, and cytokine determination in the tumor tissues and tumor-adjacent tissues before or after the injection (Supplementary Figure 9). Despite the visible tissue damage caused by in-situ vesicle injection, there were no significant signs of tissue swelling or changes in inflammation cytokines levels. These results strongly suggest that inflammation is not the primary factor responsible for the detachment of PEG₂₀₀₀ segments from NVs.

As suggested, we conducted additional safety experiments to assess serum cytokine level and leukocyte ratio in blood, spleen, and liver, following sgPik3cg-DHP/DGA-NVs or OMV treatment. The administration of sgPik3cg-DHP/DGA-NVs (1×10^{10} per mice, about 94 μ g in total protein amount) derived from protoplasts rapidly restored inflammatory factors and neutrophil proportions to their baseline levels (Figure 2g and Supplementary Figure 10e-g). In contrast, following the treatment of OMVs (dose: 1×10^{10} per mice, about 10 μ g in total protein amount), mice still maintained elevated levels of serum cytokines and neutrophil ratio in blood, spleen, and liver (Figure 2g and Supplementary Figure 10e-g). Moreover, mice subjected to OMV injection experienced fatalities whereas all mice treated with sgPik3cg-DHP/DGA-NVs survived (Supplementary Figure 10d). These findings provide compelling evidence to support the enhanced safety profile of sgPik3cg-DHP/DGA-NVs.

- 4. There is no data in the paper showing pathway modulation by the KO of PIK3cg, or the induction of TLR9 or other inflammatory signalling pathways in the tumour, normal cells or macrophages. All the data are by inference with outcome biomarkers such as CD206/CD86, or cytokine production being taken as indicative of pathway modification. Evidence of signalling changes (PI3Kg associated, TNF, IFN, IL6 etc.) should be provided in the in vitro experiments. Modification of PI3Kg signalling should be compared to IPI549.**

Response: As suggested, we performed western blot analysis to examine signaling pathway related markers (PI3K γ : p-AKT, p-C/EBP β and p-p65; TLR9: p-IRAK4; TNF: p-TAK1; IFN: p-STAT1 and IL-6: p-STAT3) in M2-BMDMs after NVs treatment as previous literatures (*Nature*. 2016;539(7629):437-442; *J*

Biol Chem. 2018;293(39):15195-15207; *J Biol Chem.* 2005;280(8):7359-68; *EMBO J.* 2001;20(1-2):91-100; *PNAS.* 2013;110(42):16975-80.) (Figure 5d and Supplementary Figure 20). Gene editing of *Pik3cg* by *sgPik3cg*-DHP/DGA-NVs led to a substantial reduction of p-AKT and p-C/EBP β levels and an elevation of p-p65 level, demonstrating more potent inhibitory effects when compared to the IPI549-treated group (Figure 5d). Furthermore, the efficiency of PIP2 conversion to PIP3, which is a substrate of PI3K γ , was significantly diminished in macrophages treated with *sgPik3cg*-DHP/DGA-NVs (Figure 5e), consistent with the western blot results in Figure 5d. This activation of the TLR9 pathway by *sgPik3cg*-DHP/DGA-NVs, resulting in elevated p-IRAK4 levels in the *sgPik3cg*-DHP/DGA-NVs-treated group compared to the IPI549 group (Figure 5d). The stimulatory effects of NVs were partially impaired by the treatment of the TLR9 antagonist ODN2088. Moreover, vesicle stimulation led to the upregulation of pro-inflammatory factors such as TNF- α , IFN- γ and IL-6 (Figure 5g), resulting in significant increases in p-TAK1, p-STAT1, and p-STAT3 levels (Supplementary Figure 20).

- 5. The in vivo reduction of PI3K γ protein appears very modest from the data shown. Does this imply there is not good modification of PI3K γ across the macrophage population? The poor modification of PI3K γ protein levels may be enough but a comparison with IPI549 modification of monocyte / macrophage function through PI3K γ signalling should be performed to establish the relevance of the effect. This is important as there is an impact of the TLR signalling and the balance of TLR vs PI3K γ in the context of the therapeutic effect of the NV is not clear throughout the studies. The authors have tried to address this with the use of a TLR antagonist compound but it does not provide enough data to determine whether modification of the PI3K γ mediated differentiation program is a major or minor driver of the effect.**

Response: A single administration of *sgPik3cg*-DHP/DGA-NVs yielded an indel rate of 15.3% within TAMs (Figure 6k-l). However, with multiple administrations, the indel rate in TAMs reached 34.4% (Supplementary Figure 24f-g). Western blot analysis demonstrated a significant downregulation of PI3K γ and related downstream signaling molecules, p-AKT and p-C/EBP β , in TAMs from the *sgPik3cg*-DHP/DGA-NVs treatment group (Figure 7g). The conversion

efficiency of PI3K γ substrate PIP2 to PIP3 was markedly reduced in the *sgPik3cg*-DHP/DGA-NVs treated group, exhibiting even greater inhibitory efficacy compared to the IPI549-treated group (Figure 7f). These findings clearly indicate that *sgPik3cg*-DHP/DGA-NVs significantly inhibit the PI3K γ pathway in TAMs.

To further confirm the role of TLR9 pathway activation in *sgPik3cg*-DHP/DGA-NVs-mediated antitumor efficacy, we conducted experiments using TLR9 gene knockout mice besides employing a TLR9 antagonist (Supplementary Figure 27 and 28). The therapeutic efficacy of *sgPik3cg*-DHP/DGA-NVs was significantly diminished in TLR9 knockout mice compared to wild-type mice, resulting in therapeutic effects similar to those of IPI549. The results of TAMs phenotypic transformation ratio, subsequent T cell activation and secretion of inflammatory cytokines were consistent with the therapeutic outcomes (Supplementary Figures 27 and 32). Furthermore, given the comparable therapeutic effects between the *sgControl*-DHP/DGA-NV combined with IPI549 treatment group and *sgPik3cg*-DHP/DGA-NVs treated mice (Figure 7), these results strongly suggest that TLR9 pathway activation and PI3K γ pathway blockade hold equally pivotal roles in the therapeutic efficacy of *sgPik3cg*-DHP/DGA-NVs.

- 6. Where IPI549 is used as a comparator in vivo dosing should be maintained compared to the NV. For example in Figure 5 it seems that IPI549 was only dosed for 6 hours before cells were transferred to new media, the legend that the cells were then cultured without IPI549. A fair comparator would be to continue the incubation with IPI549 following transfer to new media.**

Response: As suggested, IPI549 was employed for sustained treatment in both cellular and murine models (Figures 5d-h and 7, Supplementary Figures 21, and Supplementary Figures 27). The findings reveal that *sgPik3cg*-DHP/DGA-NVs treatment outperformed the IPI549-treated group in terms of macrophage repolarized ratio and therapeutic efficacy.

- 7. In Figure 6 the legend implies IPI549 was given as one dose and not continually through the 48hr time course of the experiment. Experiments should be repeated with continuous IPI549 as a control. The rendering of the fluorescence images in Panel b do not reflect the quantification in the graph. It is also surprising to see such marked differences in biodistribution to liver**

and tumour using this approach (particularly with repeated dosing). The images of the tumour distribution in the right hand Panel of B appear odd compared to other panels. It would be expected to see more in the tumour capsule, does the large red signal in the tumour indicate presence of necrosis? Shown more detailed images would be aid interpretation of this and other similar data. Some images are shown are Figure 7 panel but these are only selected fields and the IHC has not been extensively quantitated.

Response: As suggested, we administered IPI549 continuously via intragastric administration as the control group, and the treatment outcomes are depicted in Figure 7-8 and Supplementary Figure 26-27 and 30-32. These data indicate the better therapeutic efficacy of *sgPik3cg*-DHP/DGA-NVs compared to the IPI549-treated group.

As suggested, we conducted a thorough review of the original images and we sincerely apologize for the inadvertent use of images from mice that received different dosages which led to the inconsistencies in vesicle distribution and the lack of variation in radiant efficiencies in both the *in vivo* live mouse imaging and *ex vivo* organ fluorescence images. In the revised version, we conducted a thorough reevaluation of vesicle distribution, employing the same cohort of mice for both live imaging and organ imaging studies. The current findings (Figure 6a-b) unequivocally illustrate a substantial accumulation of *sgPik3cg*-DHP/DGA-NVs within the tumor site, whereas the *sgPik3cg*-NV and *sgPik3cg*-DGA-NV treated groups exhibit modest vesicle accumulation after a single administration of nano vesicles. Notably, the vesicle fluorescence intensity within the tumor tissue of mice treated with *sgPik3cg*-DHP/DGA-NVs exceeds that of the liver (Figure 6b).

To further quantify vesicle biodistribution, we collected tissues from various organs, homogenized them, and conducted fluorescence quantification analysis (Figure 6d and Supplementary Figure 22a). The results of fluorescence quantification align with the organ imaging analysis, clearly demonstrating a significant enrichment of fluorescently labeled nano vesicles within the tumor tissue of mice treated with *sgPik3cg*-DHP/DGA-NVs. Moreover, the fluorescence intensity within the tumor tissue from mice treated with *sgPik3cg*-DHP/DGA-NVs was higher than that in the liver tissue. Fluorescence images in Figure 6e show the presence of red fluorescently labeled NVs within the tumor tissue of mice treated with *sgPik3cg*-DHP/DGA-NVs. Further localization

analysis revealed significant co-localization of red fluorescence (Cy5-labeled vesicles) and green fluorescence (F4/80-stained macrophages), reinforcing the uptake of these vesicles by TAMs (Figure 6e-f). Flow cytometry of intratumoral cells treated with Cy5-labeled *sgPik3cg*-DHP/DGA-NVs showed a substantially higher rate of Cy5 positivity in TAMs compared to other cell types (Supplementary Fig. 22b). These findings indicate that the prominent red fluorescence within the tumor is attributable to the accumulation of labeled vesicles in TAMs.

The quantification results of immunohistochemical staining (Figure 8b in previous version) were shown in Supplementary Figure 30a.

- 8. Is there editing of cells in the bone marrow or peripheral blood? There is little data showing editing of cells in other organs e.g. macrophage from the liver, spleen bone marrow. Are other cell types edited e.g. epithelial cells, endothelial cells.**

Response: We employed flow cytometry to detect Cy5 fluorescent-positive cells in relevant organs, including bone marrow, peripheral blood, tumor, liver, and spleen, followed by the sorting of the corresponding cell populations for T7E1 analysis. Bone marrow cells consistently displayed Cy5-negative fluorescence (data shown below). As shown in Supplementary Figure 22b-c, the peak ratio of Cy5⁺ macrophages in the liver, spleen and tumors were 11.11%, 10.65%, 50.63%, respectively. In peripheral blood, the peak proportion of Cy5-positive cells emerged approximately 6 hours after *sgPik3cg*-DHP/DGA-NVs injection, with the highest Cy5-positive percentage of immune cells (monocytes) remaining below 10%. The indel rate in peripheral blood immune cells at 24 hours was merely 3.6%. In the liver, the highest Cy5-positive cell populations were found in monocytes, hepatocytes, and endothelial cells, constituting roughly 14% of the total, with an editing efficiency approaching 3.2%. The highest ratio of Cy5⁺ cell population in the spleen was macrophage (10.65%), with an indel rate of 2.8%. In comparison to the TAMs' indel rate of 15.3% following a single administration and 34.4% after multiple administrations, the present findings suggest that *sgPik3cg*-DHP/DGA-NVs achieve a more precise editing of TAMs (Figure 6k-l and Supplementary Figure 24g).

Figure legend: The ratios of Cy5-positive cells in bone marrow cells of 4T1 tumor-bearing mice in vein injected with 1×10^{10} *sgPik3cg*-DHP/DGA-NVs were examined by flow cytometry at indicated time points. n = 3 mice per time point.

9. **There is little broader immune cell biomarker data shown apart from CD8 accumulation Figure 8 (where the duration of treatment is not stated). It is not clear what the effect of the treatments are on other cells, particularly immune cells. For example is there a change in the neutrophils in the peripheral blood or tissues (neutrophils also depend on PI3Kg for activation). More comprehensive biomarker assessment would be important given the dual MOA of the formulation.**

Response: As suggested, we conducted flow cytometric analysis to evaluate alterations in the proportions of immune cell populations within tumor tissues and peripheral blood following treatment, along with the percentages of activated T cells and DC cells. The results in Figure 8e and Supplementary Figure 30b-c and 32a unequivocally illustrate a significant ratio of CD4⁺ and CD8⁺ T cells, accompanied by heightened expression of activation and proliferation markers (IFN- γ ⁺, Ki67⁺, and Granzyme⁺), specifically within tumor tissues after *sgPik3cg*-DHP/DGA-NVs treatment. Conversely, there is a reduction in the proportion of monocytes and an upregulation of DC ratio, while the proportions of neutrophils and other cell populations exhibit no statistically significant alterations. The flow cytometric results from peripheral blood closely mirror the trends observed in 4T1 tumor tissues, featuring an augmentation in the proportions of CD4⁺ and CD8⁺ T cells, coupled with a decrease in monocytes, while the proportions of other cell populations remain unaltered (Supplementary Figure 31).

Moreover, we conducted transcriptome sequencing analysis, employing methodologies encompassing GO and GSEA analysis, which unveiled that *sgPik3cg*-DHP/DGA-NVs treatment triggers T cell activation, immune cell activation and toll-like receptor pathways (Figure 8a-c). By analyzing these flow cytometry and transcriptome data, we postulate that *sgPik3cg*-DHP/DGA-NVs repolarized TAMs toward via the activation of the TLR9 pathway and blockade of the PI3K γ pathway. Consequently, this leads to an enhanced secretion of factors such as TNF- α , IFN- γ and IL-12, concomitant with a reduction in immunoregulatory factors like TGF- β and IL-10 (Figure 8d). These alterations foster DC and T cell activation and proliferation, thereby bestowing an anti-tumor effect.

- 10. A combination experiment with a PD1 inhibitor should be performed to confirm claims of the potential to enhance immunotherapy. This should be benchmarked versus IPI549.**

Response: As suggested, we conducted supplementary treatment experiments involving the co-administration of a PD-1 inhibitor and *sgPik3cg*-DHP/DGA-NVs (Supplementary Figure 35). The outcomes demonstrated that the combined treatment group exhibited superior efficacy compared to the individual PD-1 inhibitor and *sgPik3cg*-DHP/DGA-NVs treatment groups. Notably, its therapeutic effectiveness was comparable with that of the PD-1 inhibitor and IPI549 co-treatment group. These findings suggest that the therapeutic approach of combining PI3K γ blockade with PD-1 inhibitors holds significant promise in the field of clinical immunotherapy.

- 11. Does depleting CD8 T-cells block efficacy in the 4T1 or CT26. This is an important control to confirm the mode of action as being T cell mediated.**

Response: According to the reviewer's suggestion, CD8 T cell deletion greatly impaired the therapeutic effect of *sgPik3cg*-DHP/DGA-NVs, indicating the therapeutic effects of NVs were partially dependent on the activation of CD8⁺ T cells (Supplementary Figure 34).

- 12. Efficacy studies are performed with repeated administration. Does blocking TNF or IL6 have an impact on efficacy achieved with repeated administration? Are dendritic cells activated with repeated administration of NV?**

Response: The results from transcriptome sequencing and ELISA analysis

demonstrated a notable elevation in TNF- α levels within mouse tumor tissues following treatment with sgPik3cg-DHP/DGA-NVs (Figure 8d). Meanwhile, the levels of IL-6 remained unaltered, as depicted in Figure 8d. Given the cytotoxic impact of heightened TNF- α levels on tumor cells over a short timeframe, we employed neutralizing antibodies targeting TNF- α , which resulted in a reduction in the therapeutic efficacy of sgPik3cg-DHP/DGA-NVs, as observed in Supplementary Figure 33.

Furthermore, there was a significant increase in the proportion of CD80⁺ and CD86⁺ DC cells within mouse tumor tissues after sgPik3cg-DHP/DGA-NVs treatment, indicative of DC cell activation (Supplementary Figure 30c). These findings suggest that the TAM phenotype reversal facilitated by sgPik3cg-DHP/DGA-NVs leads to a transition of the entire tumor microenvironment from a cold to a hot state, which enhances the efficacy of T cell-mediated anti-tumor effects.

13. A control experiment of continuous IPI549 treatment with the sgControl – DHP/DGA-NV should be performed to baseline the efficacy derived from the contribution of PI3K γ + the independent NV constituents to treatment.

Response: As suggested, sgControl-DHP/DGA-NVs and IPI549 were used to treat 4T1 tumor bearing mice. Results in Fig. 7 indicate the similar therapeutic outcomes observed between the group receiving sgControl-DHP/DGA-NVs in combination with IPI549 treatment and the group treated with sgPik3cg-DHP/DGA-NVs. These findings strongly indicate that the activation of the TLR9 pathway and the blockade of the PI3K γ pathway play equally essential roles in the therapeutic effectiveness of sgPik3cg-DHP/DGA-NVs.

14. What fraction of cells are edited in each in vivo administration? How long do edited cells persist for? Is the editing short lived? A time course is required to determine the utility of the approach. This time course should incorporate normal and tumour tissues.

Response: We employed flow cytometry to detect Cy5-positive cells in key organs such as the bone marrow, peripheral blood, tumor, liver, and spleen. Cells from these regions were then isolated for T7E1 analysis. A single vesicle administration resulted in a peak of 50.63% Cy5-positive TAMs at 72 hours, with the proportion dropping to 6.62% after 10 days. Correspondingly, the indel rate peaked at 15.3% at 72 hours and stabilized at around 2.5% after 10 days

post-single dose of sgPik3cg-DHP/DGA-NVs (Figure 6i-l). With six cycles of vesicle administration, Cy5-positive TAMs surged to 73.64%, achieving a TAM editing efficiency of 34.4% (Supplementary Fig. 24e-g).

We concurrently evaluated the rates of Cy5-positivity and editing efficiency across specific cell populations in the bone marrow, blood, liver, tumor, and spleen. As depicted in Supplementary Figure 22b-c, hepatocytes, liver monocytes and endothelial cells displayed the highest Cy5-positivity, constituting around 14% of the total population, with an editing efficiency nearing 3.2%. At 72 hours, aside from tumor cells and various liver cell populations that mostly exhibited sub-3% editing rates, editing in other organs was negligible. These results underscore the precision of TAM editing achieved with sgPik3cg-DHP/DGA-NVs.

15. Is there accumulation of the NV over time with the repeated administration? No PK is shown.

Response: As suggested, we conducted repeated administrations of sgPik3cg-DHP/DGA-NVs and monitored both the fluorescence accumulation in the tumor site and the presence of Cy5⁺ TAMs. The outcomes, illustrated in Supplementary Figure 24, reveal a progressive increase in tumor volume following each administration. The positivity rate of TAMs reached approximately 73.64% (Supplementary Figure 24e-f). Additionally, we evaluated the fluorescence intensity in peripheral blood. After each administration, the fluorescence intensity of NVs in the plasma declined to near-zero levels within 48 hours. Subsequent administrations resulted in a recovery of blood fluorescence intensity, with each peak slightly surpassing the previous one.

16. A time course of modification of PI3Kg signalling versus activation of other pathways should be shown. For example in the experiment shown in Figure 7 what is the extent of pathway modulation for TNF, STAT, IFN signalling etc. in addition to downstream impact on PI3Kg mediated pathways.

Response: As suggested, we performed western blot analysis to examine signaling pathway related markers (PI3K γ : p-AKT, p-C/EBP β and p-p65; TLR9: p-IRAK4; TNF: p-TAK4; IFN: p-STAT1 and IL-6: p-STAT3) in TAMs after NVs treatment at indicated time points (Figure 7g and Supplementary Figure 26c). sgPik3cg-DHP/DGA-NVs treatment led to the gradually downregulation of PI3K γ , p-AKT and p-C/EBP β , and the upregulation of p-IRAK4, p-p65,

p-STAT1 and p-TAK1, which demonstrated the impairment of PI3K γ pathway and the activation of TLR9, TNF- α and IFN- γ signaling pathway. Nevertheless, in contrast to the results observed in *in vitro* cell experiments, no discernible alterations were detected in IL-6 and p-STAT3 within TAMs following NVs treatment. Transcriptome sequencing analysis revealed a multitude of genes displaying aberrant changes, implicating pathways including TNF- α , IFN- γ , T cell activation, NF- κ B activation, and TLR-related pathways (Figure 8a-b). Further validation through GSEA analysis solidified the positive association between these pathways and the effects of *sgPik3cg*-DHP/DGA-NVs treatment (Figure 8c and Supplementary Figure 29).

These comprehensive findings strongly indicate that *sgPik3cg*-DHP/DGA-NVs, when exerting therapeutic effects *in vivo*, effectively reverse the tumor-associated macrophage phenotype through a dual mechanism involving the inhibition of the PI3K γ pathway and the activation of the TLR9 pathway. This secretion of factors, including TNF- α , IFN- γ , and IL-12, not only stimulates pertinent pathways within TAMs but also gradually enhances the proliferation and activation of T cells (Figure 8e), thereby contributing to the anti-tumor effect.

Reviewer 4:

This is an excellent manuscript detailing the development of a new *in vivo* Crispr tool. Using bacterial protoplasts bearing a Cas9-sgRNA RNP, the authors deliver Crispr components first to cells and then to tumors. Taken up by macrophages, the protoplasts transfect macrophages with PI3K γ siRNA. This alters macrophages by stimulating expression of pro-inflammatory factors that lead to inhibition of tumor growth.

This work is a significant development in the delivery of Crispr to cells *in vivo* and a rewarding demonstration that PI3K γ inhibition remains a valuable way to suppress tumor growth.

Response: We appreciate your time in reviewing our manuscript and offering excellent comments on our work.

REVIEWER COMMENTS

Reviewer #1 (Remarks to the Author):

1 For the components analysis of sgPik3cg-DHP/DGA-NVs issue (Figure 3-e), the ratios of groups were different. In my opinion, the consistency of protein concentrations at different groups were the basis of the experiments and determines the final results of the experiments. The author said that “the results of Supplementary Figure 13 reveals that the abundance of classic outer membrane proteins and periplasmic proteins (e.g., OmpA and DcrB) in sgPik3cg-DHP/DGA-NVs was lower than those in OMVs, consistent with the intra-group abundance ranking results”, but from Figure S13-e, it could not obtain any results about these, the author should a clear explain.

2 For the Supplementary Fig. 9 issue, it was in vitro safety assessments after treatment with different NVs. But in manuscript, it said that “importantly, this administration did not result in noticeable erythema, inflammatory cell infiltration, or cytokine upregulation (such as IL-6 and IL-1b”, The content of the two sides is inconsistent, the author should check them carefully.

3 For the characterization of polymers issue (Figure S-2 and S-3), the ¹H NMRs were not publishable. Too many noises and impurity were in the images. The author should pass PD-10 column to purity them and then do ¹H NMRs.

Reviewer #2 (Remarks to the Author):

The author addresses all issues.

Reviewer #3 (Remarks to the Author):

The authors have performed alot of work to address the comments. The data in the current version makes it much easier to follow the concepts.

Minor points to help clarity:

1. The work they have done demonstrates efficacy in many individual repeats. In the discussion can some comment be made on the specific variability in anti-tumour activity seen across the individual 4T1 studies in particular. Is the effect variable in a range of 40% - 70% TGI or is the variability low between studies. Also it would add to robustness saying how many individual 4T1 studies efficacy was demonstrated
2. The concentrations / doses for in vitro and in vivo experiments should be clear in all the figure legends.
3. The effects of IPI549 in these experiments if less than that normally seen with PI3K antagonists, this should be commented on in the discussion or results.
4. I may not have spotted this but while there are distribution studies performed on other organs, I couldn't spot and PD biomarker data to see if there was a modification of inflammatory biomarkers in "normal" tissues. this caveat should be highlighted if that data is not shown.

The questions raised by the reviewers are responded point by point as follows:

Reviewer 1:

- 1. For the components analysis of sgPik3cg-DHP/DGA-NVs issue (Figure 3-e), the ratios of groups were different. In my opinion, the consistency of protein concentrations at different groups were the basis of the experiments and determines the final results of the experiments. The author said that “the results of Supplementary Figure 13 reveals that the abundance of classic outer membrane proteins and periplasmic proteins (e.g., OmpA and DcrB) in sgPik3cg-DHP/DGA-NVs was lower than those in OMVs, consistent with the intra-group abundance ranking results”, but from Figure S13-e, it could not obtain any results about these, the author should a clear explain.**

Response: We fully agree with the reviewer’s observation and appreciate the opportunity to address the issues raised concerning Supplementary Fig. 13 from our previous revision. In this revised version, we have updated the sections on supplementary methods and results (Fig. 3e-f, Supplementary Fig. 13). Firstly, we ensured that the quantities for different samples in the LC-MS analysis were consistently maintained at 100 µg each. We then employed MaxQuant software to analyze the protein mass spectrometry data. The iBAQ protein intensity, as outlined in the literatures (*Nature* **473**, 337-342 (2011); *Mol Cell Proteomics* **11**, M111.014050 (2012)), was applied to rank the abundance of distinct proteins within each group and the top 200 abundant proteins were shown in Supplementary Table 4-7. The subcellular localization of the top 200 abundant proteins in each sample was analyzed (Fig. 3e). Compared to the types found in OMVs, the *sgPik3cg-DHP/DGA-NVs* contained fewer kinds of outer membrane and periplasmic proteins. Additionally, we compared the outer membrane proteins and periplasmic proteins across OMVs and *sgPik3cg-DHP/DGA-NVs* utilizing LFQ intensity, represented by a normalized intensity profile generated using a specific algorithm (*Mol Cell Proteomics* **13**, 2513-2526 (2014); *Nat Protoc* **11**, 2301-2319 (2016)). Protein LFQ intensities were log₂-transformed before heat mapping to reduce distributional skew and to give approximate normality, as reported previously (*Transl Psychiatry* **6**, e959 (2016)). The results demonstrated that the abundance of certain classical outer membrane proteins and periplasmic proteins in *sgPik3cg-DHP/DGA-NVs* was indeed lower than those in OMVs, as illustrated in Supplementary Fig. 13.

- 2. For the Supplementary Fig. 9 issue, it was in vitro safety assessments after treatment with different NVs. But in manuscript, it said that “importantly, this administration did not result in noticeable erythema, inflammatory cell infiltration, or cytokine upregulation (such as IL-6 and IL-1b”, The content of the two sides is inconsistent, the author should check them carefully.**

Response: We thank the reviewer’s insightful feedback. To improve clarity and readability of our study, we have reorganized the presentation of relevant results in the Results section and provided clearer explanations for the corresponding experiments. The details in the revised section are as follows: “Moreover, to examine the stability of the hydrazone bond of DHP in 4T1 tumor tissue, we performed *in situ* injections of RhB-DHP/DGA-CFSE-labeled NVs into both tumor and adjacent tissue sites. Distinct localization of CFSE-labeled NVs (green) from the RhB-labeled PEG₂₀₀₀ was observed in the acidic tumor tissue. However, the fluorescence signals of CFSE and RhB completely overlapped in the tumor-adjacent tissue due to its neutral pH environment (Fig. 2f). Crucially, these injections did not induce noticeable erythema, inflammatory cell infiltration, or upregulation of pro-inflammatory cytokines such as IL-6 and IL-1 β (Supplementary Fig. 9), indicating that inflammation is not the primary factor responsible for PEG₂₀₀₀ segments detachment from NVs. This also implies that *sgPik3cg*-DHP/DGA-NVs exhibit favorable biocompatibility and safety. In addition to these *in situ* vesicle injections, we conducted a comprehensive *in vitro* and *in vivo* safety evaluation of *sgPik3cg*-DHP/DGA-NVs, including analyses of LPS content, cytotoxicity, hemolytic toxicity, and evaluation of inflammatory factors in serum and immune cell populations in organs from mice following systemic administration of NVs. OMVs, naturally secreted by bacteria with known high toxicity, were utilized as controls in these experiments.”

- 3. For the characterization of polymers issue (Figure S-2 and S-3), the 1H NMRs were not publishable. Too many noises and impurity were in the images. The author should pass PD-10 column to purify them and then do 1H NMRs.**

Response: In accordance with the suggestion, we have further purified the two polymers (DHP and DGA) and conducted proton nuclear magnetic resonance (¹H NMR) analysis. The updated ¹H NMR spectra for DHP and DGA are now presented in Supplementary Fig. 2b and Supplementary Fig. 3b.

Reviewer 2:

The author addresses all issues.

Response: We appreciate your time and effort you have dedicated to reviewing our manuscript and offering insightful comments on our work.

Reviewer 3:

- 1. The work they have done demonstrates efficacy in many individual repeats. In the discussion can some comment be made on the specific variability in anti-tumour activity seen across the individual 4T1 studies in particular. Is the effect variable in a range of 40% - 70% TGI or is the variability low between studies. Also it would add to robustness saying how many individual 4T1 studies efficacy was demonstrated.**

Response: Thank you for providing helpful and detailed suggestions. In our study, we evaluated the therapeutic efficacy of *sgPik3cg*-DHP/DGA-NVs in four independent 4T1 mouse tumor models. The observed tumor growth inhibition (TGI) ratio for *sgPik3cg*-DHP/DGA-NV ranged from 70.5% to 81.8%. These results indicate a consistently low variability in the anti-tumor effects mediated by *sgPik3cg*-DHP/DGA-NV, underscoring its promising potential in cancer immunotherapy. We have incorporated these observations and relevant discussions into the discussion section of our manuscript.

- 2. The concentrations / doses for in vitro and in vivo experiments should be clear in all the figure legends.**

Response: We are grateful for the reviewer's suggestion. We have thoroughly revised the manuscript to ensure that the concentration and doses are clearly stated in all figure legends for both *in vitro* and *in vivo* experiments.

- 3. The effects of IPI549 in these experiments if less than that normally seen with PI3Kg antagonists, this should be commented on in the discussion or results.**

Response: In our study, IPI549 showed a tumor growth inhibition (TGI) ratio of 39.5% to 51.5% in the 4T1 model from two experiments, and 45.1% in the MC38 model from one experiment, consistent with results from prior research (TGI ratios

of 19-38% in 4T1 tumor model and 40-45% in MC38 tumor model) (*Nature* **539**, 443-447 (2016)). Additionally, we analyzed the TGI ratios of other PI3K γ antagonists (TG100-115, ZX-4081, IPI145, and AZD3458) based on tumor growth curves from previous reports (*Adv Sci (Weinh)* **6**, 1900327 (2019); *Cancer Res* **81**, 1384-1384 (2021); *Circulation* **138**, 696-711 (2018); *Mol Cancer Ther* **20**, 1080-1091 (2021)). Those antagonists exhibited TGIs ranging from 21.2% to 44.1% in the 4T1 tumor model and around 40.0% in the MC38 tumor model. Notably, the tumor-suppressive efficacy of sgPik3cg-DHP/DGA-NVs surpasses that of standard PI3K γ antagonists. We have included these observations in the discussion section for comprehensive analysis.

- 4. I may not have spotted this but while there are distribution studies performed on other organs, I couldnt spot and PD biomarker data to see if there was a modification of inflammatory biomarkers in "normal" tissues. this caveat should be highlighted if that data is not shown.**

Response: Thank you for pointing out this aspect. In our *in vivo* distribution experiments, sgPik3cg-DHP/DGA-NVs treatment led to the gene editing in a small fraction of cells in normal organs, such as the liver and spleen. Although we did not specifically assess PI3K γ protein expression levels and related inflammatory factors in these normal organs post-NV administration, our safety assessments indicated that immune cell populations in the liver and spleen reverted to baseline levels 24 hours after a single dose of sgPik3cg-DHP/DGA-NV (as seen in Supplementary Fig. 10 f-g). Furthermore, organs and serum collected from mice subjected to multiple doses of sgPik3cg-DHP/DGA-NVs for H&E staining and biochemical analysis. The H&E staining results showed no significant immune cell infiltration or tissue damage in organs of mice treated with sgPik3cg-DHP/DGA-NVs (Supplementary Fig. 25g). Biochemical indexes including ALT and AST (liver function), BUN and CR (kidney functions), and LDH (heart function), in the serum of treated mice were comparable to those in the control group (Supplementary Fig. 25b-f). These findings imply that nonspecific editing did not lead to significant inflammation or tissue damage in normal organs, underscoring the safety of our vesicle-based

therapeutic approach. We plan to investigate whether sg*Pik3cg*-DHP/DGA-NVs mediated-gene editing in normal organs might activate tissue-resident immune cells and elicit inflammatory responses in future studies. The related content has been added to the discussion section of the revised version.

REVIEWERS' COMMENTS

Reviewer #1 (Remarks to the Author):

This manuscript reported by Junfeng Zhang et al “Bacterial Protoplast-Derived Nanovesicles Carrying CRISPR Cas9 Tools Efficiently Re-educate Tumor-Associated Macrophages for Cancer Immunotherapy”. The authors had modified all the points by comments, and in this style it could consider publication, following minor points should be answered.

1 In “biodistribution assay of NVs in mouse tumor models and Anti-tumor activity of sgPik3cg-DHP/DGA-NVs” sections, the author noted that “ we administered 100 μ L of PBS containing 1×10^{10} Cy5-labeled sgPik3cg-NVs.....”, how about the units all of them? Because this manuscript involved a large of animal experiments, the dose and sequence of administration were very important. The author should carefully explain.

The questions raised by the reviewers have been addressed as follows:

Reviewer 1:

This manuscript reported by Junfeng Zhang et al “Bacterial Protoplast-Derived Nanovesicles Carrying CRISPR Cas9 Tools Efficiently Re-educate Tumor-Associated Macrophages for Cancer Immunotherapy”. The authors had modified all the points by comments, and in this style it could consider publication, following minor points should be answered.

1 In “biodistribution assay of NVs in mouse tumor models and Anti-tumor activity of sgPik3cg-DHP/DGA-NVs” sections, the author noted that “ we administered 100 µL of PBS containing 1×10^{10} Cy5-labeled sgPik3cg-NVs.....”, how about the units all of them? Because this manuscript involved a large of animal experiments, the dose and sequence of administration were very important. The author should carefully explain.

Response: We appreciate the detailed and insightful feedback provided. To determine the most effective dosing and administration protocol for our study, a range of doses and timings were evaluated in preliminary trials. The chosen regimen, administering 1×10^{10} NVs per mouse bi-daily, was based on these initial findings. This specific dosage and frequency were found to optimize targeting efficiency and editing capability in tumor-associated macrophages (TAMs). Each 1×10^{10} NV dose contains 20.62 µg of ribonucleoprotein (RNP), aligning with established *in vivo* RNP dosing ranges reported in previous studies (referencing *Nat Commun* **11**, 3232 (2020), *Nano Lett* **21**, 9761-9771 (2021), *Adv Sci* **4**, 1700175 (2017)). We have incorporated additional details on the rationale behind our dosing regimen in the methods section of our manuscript.